# QBO modulation of stratospheric water vapour in the Asian monsoon

Cristina Peña-Ortiz[1], Nuria Pilar Plaza[2], David Gallego[1], Felix Ploeger[3,4]

[1]Universidad Pablo de Olavide, Sevilla, 41013, Spain
[2]Centro de Investigaciones sobre Desertificación, Consejo Superior de Investigaciones Científicas (CIDE-CSIC), 46113 Moncada, Valencia, Spain
[3]Institute of Climate Research, Stratosphere (IEK-7). Forschungszentrum Jülich, Jülich, 52428, Germany
[4] Institute for Atmospheric and Environmental Research, University of Wuppertal, Wuppertal, 42119, Germany

*Correspondence to*: Cristina Peña-Ortiz (cpenort@upo.es)

**Abstract.** The Asian Monsoon (AM) plays a key role in the transport of water vapour to the lower stratosphere and contributes significantly to the wet phase of the annual global stratospheric water vapour cycle. Although it is known that the

QBO is one of the main drivers of the interannual variability of the AM water vapour, the physical mechanisms responsible for this variability remain unclear. Here we have used daily MLS data for the period 2005-2020 to characterize the QBO signature on the lower stratosphere AM water vapour during the boreal summer. We show that the QBO has the strongest impact during August, when QBO-W minus QBO-E differences may reach 1ppmv at 100hPa, although a significant signature is also observed during July. We find that the region whose temperature controls the QBO signal on water vapour

over the AM differs between July and August. In July, when the key region is over the tropical Indian Ocean, the QBO modulation of the AM water vapour occurs in phase with the signal over the equator whereas in August, when the key region is at the subtropics, over the southern edge of the monsoon, the signal over the AM is opposite to that over the equator. Our results reveal that the QBO signal on the temperature on the south side of the AM anticyclone, which ultimately has an impact on AM water vapour is, in turn, modulated by the QBO impact on tropical clouds. Thus, we find that the QBO

signature on clouds over the eastern Indian Ocean gives rise to Rossby wave trains that produce variations in the circulation over the southern side of the AM anticyclone such that weaker anticyclone over this region generates an increase in water vapour and vice versa.

## 1 Introduction

Almost all of the water vapour in the lower stratosphere appears as a result of transport from the troposphere, with a small
additional contribution due to methane oxidation. This transport occurs mainly through the tropical tropopause, often call the "cold trap", where horizontal transport causes air masses to experience extraordinarily low temperatures as they pass through

the coldest regions, limiting the water vapour concentration in the lower stratosphere to a few parts per million (Holton and Gettelman, 2001; Fuegistaler et al., 2004, Fueglistaler et al., 2009; Liang et al., 2011; Randel et al., 2004; Zhou et al., 2001, 2004). However, despite its low concentration, water vapour in the lower stratosphere plays a fundamental role in radiative

balance in the tropics (Brindley and Harries, 1998) and in the ozone chemistry (Dvortsov and Solomon, 2001; Stenke and Grewe, 2005).

It is generally accepted that monsoon regions play a key role in troposphere-to-stratosphere transport and thus in controlling the concentration of water vapour in the lower stratosphere. In the Northern Hemisphere, stratospheric water vapour concentration shows two maxima between 150hPa and 70hPa over the Asian and North American monsoon regions

(Rosenlof et al., 1997; Park et al., 2007). These summer climatological maxima can influence water vapour over much of the Northern Hemisphere through their large-scale transport (Ploeger et al., 2013) and are very likely to contribute significantly to the wet phase of the annual global stratospheric water vapour cycle. In this regard, Nuetzel et al., (2019) showed, through model simulation, that the AM contributes ~15% to tropical tape recorder wet phase, and ~30% to NH extratropical lowermost stratosphere summertime water vapour maximum. Different global models suggest that convection over the

Southeast AM region represents the major source of moisture to the stratosphere (Bannister et al., 2004).

Despite their importance, there is much uncertainty regarding the physical mechanisms that generate water vapour maxima over monsoon regions. Previous studies (Park et al., 2007; Wright et al., 2011; Randel et al., 2015) have suggested that the transport of water vapor to the AM lower stratosphere is controlled by dehydration temperatures and convection mainly over the southern (cold) side of the lower stratospheric anticyclonic circulation, an extended region covering from northern India

to southwestern China. Wright et al. (2011) found that dehydration of air parcels entering the AM lower stratosphere primarily occurs over this region and, consistently, Randel et al. (2015) found that, on an intraseasonal scale, stratospheric water vapour over the AM is mainly controlled by large-scale temperature variations over the southern edge of the anticyclone forced by deep convection. However, it is still unclear how important deep convection, monsoon temperature and circulation, or in situ dehydration of air masses may be. Randel and Park (2006) show consistent fluctuations in deep

convection and water vapour content of the AM anticyclone. However, on an intraseasonal scale, the peaks in stratospheric water vapour over the monsoons do not coincide either spatially or temporally with the peaks in convective activity, suggesting that horizontal transport may play a role. In fact, and contrary to expectations, Randel et al. (2015) found that deep convection over the Asian and North American monsoons is associated with a drier stratosphere, which they explain through the cooling of the lower stratosphere produced by the convection itself. In contrast, other studies suggested a

moistening effect of overshooting convection in the AM (e.g., Khaykin et al., 2022), but with the impact on the lower stratosphere water vapor budget currently under debate (Konopka et al., 2023).

Lower stratospheric water vapour over the Asian and North American monsoons exhibits very significant interannual variability in which the Quasi-biennial Oscillation (QBO) and ENSO (El Niño Southern Oscillation) appear to dominate (Randel et al., 2015). However, the physical mechanisms responsible for this variability have been poorly investigated,

hitherto. In general terms, we know that the concentration of water vapour in the stratosphere is profoundly influenced by

global circulation patterns, particularly those affecting the tropical regions, as ENSO and the QBO, through which transport preferentially occurs. The QBO dominates the interannual variability of water vapour in the lower and middle stratosphere by modulating tropical tropopause temperatures and, in spite of the fact that its signal is relatively weak at the tropopause (± 1K), it has a significant influence on the mixing ratio of rising air into the stratosphere (Giorgetta et al., 1999; Geller et al., 2002, Tian et al., 2019). However, Randel et al. (2015) found asymmetries in the QBO signal over the Asian and North American monsoons that, a priori, are not consistent with the mechanism based on temperature modulation, since the QBO signal on temperature is zonally symmetric.

In addition, the QBO also influences deep convection processes in the tropics (Giorgetta et al., 1999, Peña-Ortiz et al., 2019), determining the other important mechanism in the transport of vapour across the tropopause. Giorgetta et al. (1999) showed that during the easterly phase of the QBO there is an intensification of convection over the East Asian and Indian monsoons leading to increased cloudiness in areas close to the tropopause. However, the effect of this modulation of convection on the transport of water vapour into the stratosphere has not yet been studied. In this paper we analize the QBO impact on water vapour in the lower stratosphere over the AM. We make use of observational data from the Microwave Limb Sounder (MLS) to quantify and describe the behaviour of this signature during July and August according to the QBO phases. Additionally, we examine the QBO signature on temperature and convection and address the question concerning the role that these variables play in the QBO signal on the water vapour over the AM.

## 2 Data and Methodology

We have analyzed observations of water vapour mixing ratio and temperature in the lower stratosphere from the Microwave Limb Sounder (MLS) aboard the NASA satellite Aura (Waters et al., 2006). For both variables we have used MLS 4.2 version (Lambert et al., 2015, Livesey et al., 2020) from which we have produced gridded daily data on 100hPa and 82 hPa pressure levels by averaging profiles inside bins with resolution of 2° latitude × 5° longitude for the period 2005-2020. Water vapour measurements have been validated in several studies and have been part of a climatological overview of the AM Anticyclone (Santee et al., 2017). The single profile precision of the MLS water vapour is 7% and 15% at 82hPa and 100hPa while accuracy is 9% and 8% respectively for these two pressure levels (Table 3.9.1 of Livesey et al. (2020)). The single profile precision and accuracy of the MLS temperature data product are shown in Table 3.22.1 of Livesey et al. (2020). The precision is 0.8 K or better in the lower stratosphere while observed biases based upon comparisons with previously validated satellite based measurements range from 0 to +1 K in the lower stratosphere.

To investigate the mechanisms behind the QBO signature on the AM water vapour, we used daily values of wind, temperature, fraction of cloud cover and total diabatic heating rate, which includes components of heating due to latent heat release, radiative and turbulent heating, from the ERA5 reanalysis (Hersbach et al., 2020) over the period 2005-2020. The ERA5 reanalysis, based on the Integrated Forecasting System (IFS), has a horizontal resolution of 31km and 137 vertical levels that extend to 0.01 hPa. For this study, the fields of wind, temperature and fraction of cloud cover obtained from this

reanalysis have been further interpolated onto a 2.5º × 2.5º longitude and latitude grid, and they are extracted from the analysis available at 37 pressure levels between 1000hPa and 1hPa. Regarding the fraction of cloud cover, the comparison performed by Yao et al. (2020) between ERA5 and data from the Moderate Resolution Imaging Spectroradiometer (MODIS) aboard the Terra and Aqua satellites showed that differences of monthly mean cloud cover between ERA5 and MODIS are mostly around or less than 5% over the tropical and subtropical regions where, additionally, the correlation coefficients between these two datasets are larger than 0.8. These results allow us to conclude that the interannual variability of cloud cover is reasonably captured by ERA5.

According to Pahlavan et al. (2021), the representation of the mean fields in the QBO is very similar in ERA5 and ERA-Interim, which has been used extensively for assessing various aspects of the QBO and has been found to be quite reliable in the tropical lower and middle stratosphere.

Together with ERA5 fraction of cloud cover, NOAA-interpolated outgoing longwave radiation (OLR) data (Liebmann and Smith, 1996) is also used to explore the QBO impact on clouds. We use the daily data interpolated onto a 2.5º × 2.5º longitude and latitude grid provided by the NOAA/OAR/ESRL PSL, Boulder, Colorado, USA, from their web site at https://www.psl.noaa.gov/.

In this study, Singapore sonde monthly zonal wind has been used to define the QBO phases at five pressure levels between 70hPa and 10hPa and for two different months separately, July and August. Each wind value corresponding to each month and level was standardized by subtracting the average value for the period of study, 2005-2020, and dividing by the standard deviation. Then, the QBO easterly (QBO-E) and westerly phase (QBO-W) at each level corresponds to those cases in which the standardized zonal wind values were below -0.5 or above 0.5. Table 1 shows the number of QBO-W and QBO-E cases obtained for each month and pressure level and Singapore sonde monthly zonal wind averaged for the years defined as QBO-W and QBO-E.

| | 10hPa | 20hPa | 30hPa | 50hPa | 70hPa |
|---|---|---|---|---|---|
| **July** | 7/6 (15.2/-19.2) | 8/6 (21.7/-20.3) | 8/7 (21.3/-18.7) | 8/6 (12.2/-14.7) | 5/5 (7.8/-13.1) |
| **August** | 7/6 (16.6/-20.3) | 9/6 (18.9/-22.6) | 9/7 (20.1/-19.9) | 9/5 (11.0/-13.4) | 7/5 (6.2/-12.4) |

**Table 1: Number of years classified as QBO-W/QBO-E together with the Singapore Sonde monthly zonal wind averages for (QBO-W/QBO-E) in m/s obtained for July and August at each pressure level for the period 2005-2020.**

In order to assess the QBO impact on the lower stratosphere water vapour over the AM, we computed QBO-W minus QBO-E differences of MLS water vapour at 100hPa and 82hPa. Although, in principle, we considered all months in which the Asian Monsoon is active, from June to September, we only detected a significant QBO signal in July and August. Therefore, we have excluded from our analysis the months of June and September, during which we did not find significant anomalies over the AM (not shown). Among all the different pressure levels we have used to define the QBO index (Table 1), we have selected those for which the water vapour differences over the AM associated with QBO phases are strongest and most

significant during July and August separately. As can be seen in figures A1 and A2, the signal over water vapour in the AM at 100hPa reaches its maximum when we take the QBO index at 10hPa for the signal in July and at 20hPa for the signal in August. The fact that it is the index defined at these upper stratospheric levels that gives rise to a stronger signal on water vapour does not mean that the physical mechanism has a direct relationship with the circulation or temperature of the QBO at these levels. However, the definition of the phase at high stratospheric levels fixes the characteristics of the QBO throughout the stratosphere including the lower stratosphere and the tropopause, where the QBO wind and temperature can have an impact on lower stratospheric water vapour. In fact, in the case of July, the water vapour signal for the QBO defined at 10hPa is practically the same but with the opposite sign to the one observed in the last row of figure A1, for the QBO defined at 70hPa. However, because the QBO signal over the zonal wind weakens in the lower stratosphere, the use of levels between 70hPa and 100hPa to define the QBO phases can be problematic and significantly reduces the number of cases. Thus, although the indices that maximize the signal are referred to upper stratospheric levels, for the analysis of the possible mechanism of the signal over water vapour, we will focus on the circulation and temperature characteristics of the QBO at 100hPa.

**3 The QBO impact on the AM water vapour and the role of temperature.**

As expected, Fig. 1 shows a QBO modulation of the water vapour over equatorial latitudes, which is a well-known response to the QBO signature on equatorial temperatures. During July, QBO-W minus QBO-E differences show a water vapour decrease up to -0.7 ppmv over the tropics (for the QBO defined at 10hPa) while during August (for the QBO defined at 20hPa) these differences stays above -0.4 ppmv. The QBO impact on the tropical water vapour is also evident at 82hPa (Figs. 1b and d), which is consistent with the existence of a QBO modulation of the water vapour tape recorder resulting from its modulation of the tropical cold point tropopause (Geller et al., 2002).

By contrast with the signal on the tropical water vapour, the QBO signals over the AM shows an opposite behaviour in July and August. Thus, while the QBO signal over the AM and the tropics are in phase in July (Figs. 1a and b), they show opposite signs during August (Figs. 1c and d). As established in previous studies, the UTLS temperature plays a key role in the control of water vapour over the AM through large scale dehydration (Wright et al., 2011 and Randel et al., 2015). Furthermore, Randel et al. (2015) found that, at intraseasonal time scales, there is a lag between temperature and its impact on the AM water vapour of around 10 days. Therefore, in order to assess the link between the QBO impact on temperature and on water vapour, we first identify those regions whose temperature exerts the greatest control over the interannual variability of the AM water vapour and the lag that characterises this link. To identify these regions we computed running correlations between the AM water vapour and the 100hPa temperature field at each grid point. For that purpose, we calculated the regional average of water vapour over the AM domain (20ºN–40°N and 40ºE–140°E) for July and August over the period 2005-2020 and the averages of the daily temperatures over 31-day running windows from June 1st to September 30th at each grid point over the same time period. Results for July show maximum correlations for the

temperature over the tropical Indian Ocean average between 16 June and 16 July (Fig. 2a). For this period of the year, significant correlations extend from the tropical Indian Ocean to southern India reaching values up to 0.8 over some regions of the Indian Ocean. For the temperature average over this region,

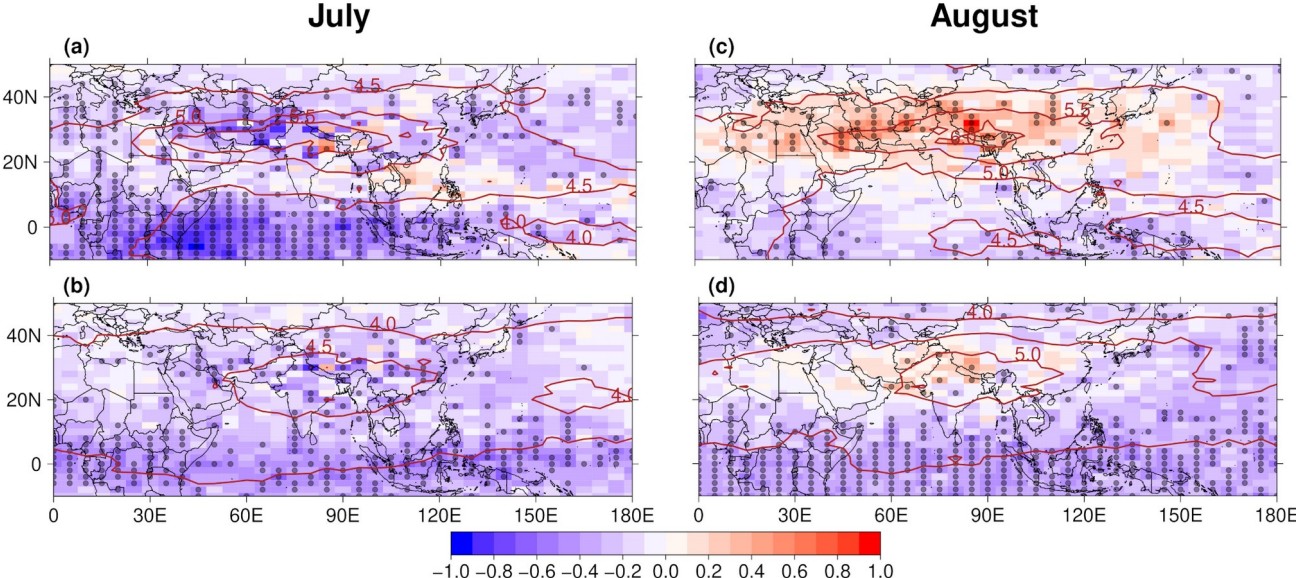

**Figure 1: QBO-W minus QBO-E differences for MLS water vapour in ppmv for July and August (left and right columns) at 100hPa (a,c) and 82hPa (b,d) over the period 2005-2020. In (a,b) differences correspond to the QBO index for July defined at 10hPa while for (c,d) we chose the QBO index defined at 20hPa for August. Dots indicate significance at the 95% confidence level. Red contours show climatological values for water vapor expressed in ppmv.**

correlations with the AM water vapour for July achieve a maximum value of 0.8 that slowly decreases as the time window moves away from the maximum correlation period, 16 June - 16 July (Fig. 2b, dark blue line). This result indicates that the temperature over the Indian Ocean is key in the control of the interannual variability of the AM water vapour during July and that its impact peaks with a time lag of about 15 days between the temperature signal and the water vapour response. It is not only in the tropical Indian Ocean that significant correlations are found. Positive correlations extend zonally over the equatorial region (not entirely shown in Fig. 3a) forming a spatial pattern consistent with the QBO signal on temperature in the tropics. By contrast, results for August show maximum correlations over the southeastern edge of the AM anticyclone, from northern India to southwestern China, which peak for the temperature field averaged between 19 July and 18 August (Figs. 2c and 2d, dark blue line), implying that there is a time lag of about 12 days between the temperature signal and the water vapour response. Furthermore, contrary to what was found for July, the AM water vapour content during August is not significatively correlated with equatorial temperatures (Fig. 2c)

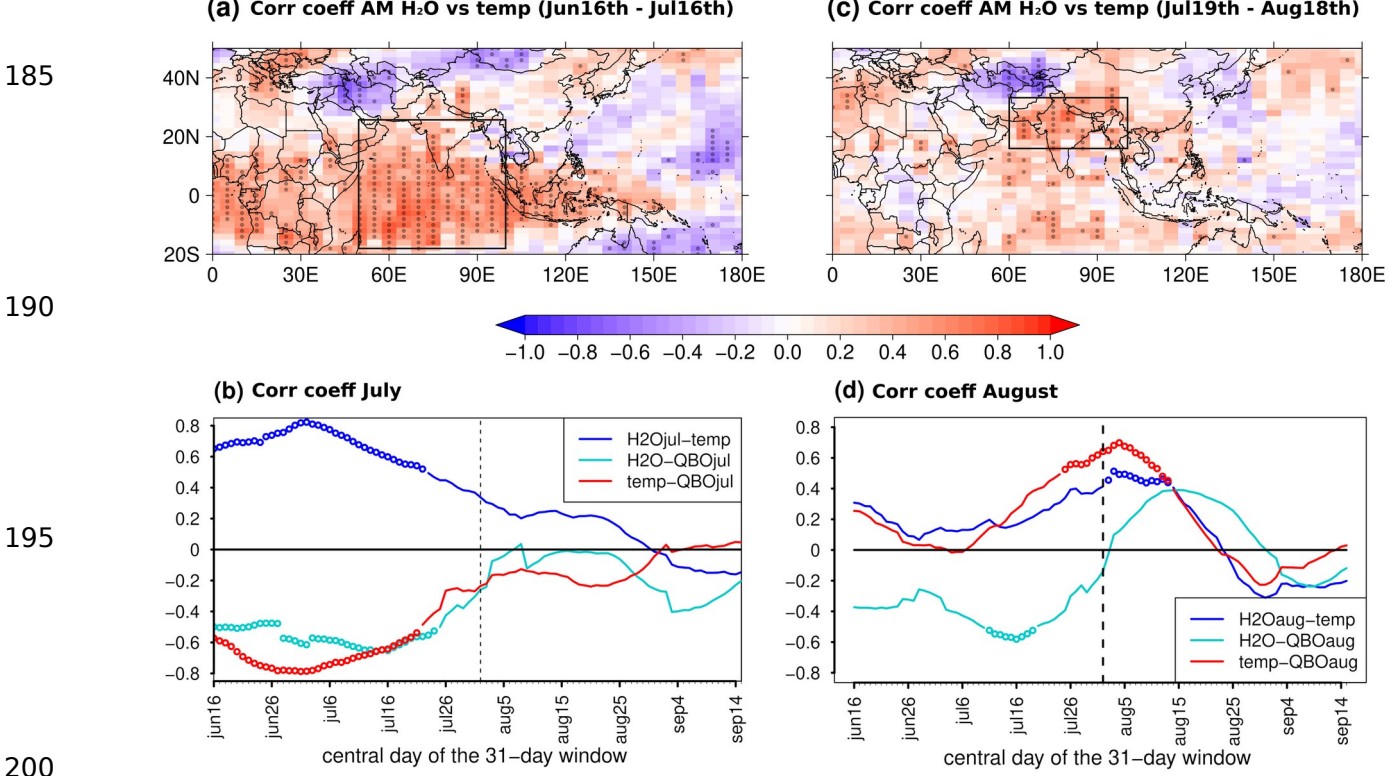

**Figure 2: (a)** Correlation between July water vapor at 100hPa over the AM [20N-40N, 40E-140E] and temperature at 100hPa at each grid point averaged between June 16th and July 16th over the period 2005-2020.**(b)** Lag sliding correlations between July AM water vapor at 100hPa and the average temperaure over the marked region in figure (a) [18S-26N, 50E-100E] and over 31-day running windows from June 1st to September 30th (dark blue line). Light blue (red) line represents lag sliding correlations between the QBO index for July at 10hPa and AM water vapor (mean temperature over the marked region in figure (a)), averaged over 31-day running windows from June 1st to September 30th. **(c)** Correlation between August water vapor at 100hPa over the AM [20N-40N, 40E-140E] and temperature at 100hPa at each grid point averaged between July 19th to and August 18th over the period 2005-2020. **(d)** Lag sliding correlations between August AM water vapor at 100hPa and the average temperaure over the marked region in figure (c) [5N-35N, 60E-100E] and over 31-day running windows from June 1st to September 30th. Light blue (red) line represents lag sliding correlations between the QBO index for August at 20hPa and the AM water vapor (mean temperature over the marked region in figure (c)), averaged over 31-day running windows from June 1st to September 30th. In (a) and (c) dots indicate significance at the 95% confidence level while in (b) and (d) circles indicate significance at the 90% confidence level.

In spite of the fact that different climate patterns may contribute to the interannual variability of the AM water vapour in the lower stratosphere (e.g., ENSO), the QBO is expected to be a major source of variability at this timescale. Thus, we make use of the spatial and temporal features of the connection between temperature and AM water vapour observed in Fig. 2 to assess the link between the QBO impact on temperature and water vapour. With this aim, Fig. 3 represents QBO related differences for the temperature at 100hPa averaged over those time windows that maximise the impact on the interannual variability of the AM water vapour during July (16 June - 16 July) and August (19 July - 18 August). For the average over the period 16 June – 16 July, QBO westerlies at 10 hPa are linked to negative temperature anomalies, when compared with the QBO easterly phase, at the equatorial UTLS (Fig. 3a) and over the tropical Indian Ocean, which is the region controlling the water vapour over the AM during this month (Fig. 2a). It should be noted that QBO westerlies defined at 10hPa correspond to the opposite phase in the lower stratosphere (for further details, see comments on Fig. 4 at the end of this section) and therefore, the observed cooling over the Indian Ocean is part of the QBO signature on the tropical tropopause temperature associated with easterly winds in the lower stratosphere. This cooling is consistent with the anomalously dry stratosphere found in July over both the tropics and the AM in Figs. 1a-b. In quantitative terms, the comparison between Figs. 3a and 1a indicates that temperature differences over the Indian Ocean between -1 K and -2 K precede a water vapour decrease of around -0.4 ppmv over the AM that can reach -0.8 ppmv to the western India. Hence, this relation between temperature and stratospheric water vapour in the monsoon region is consistent with the expected about 0.5 ppmv entry water vapour change for a 1K temperature change for globally averaged inter-annual anomalies, as found by Fueglistaler and Haynes (2005). Figures A1a and A3a in the appendix evidence the consistency between the QBO signal on the tropical UTLS temperature and water vapour in the AM for the QBO phases defined at different pressure levels.

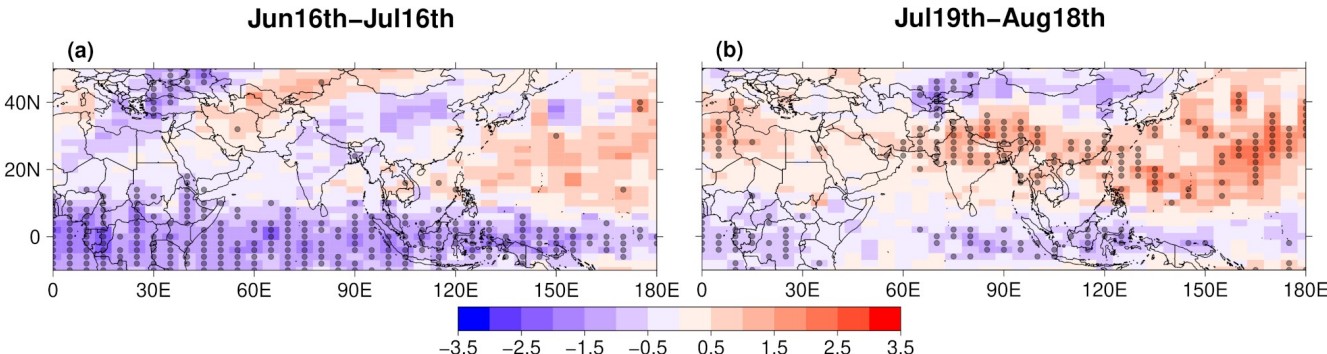

**Figure 3: QBO-W minus QBO-E differences for MLS temperature at 100hPa averaged over the 31-day period between 16 June and 16 July (a) and between 19 July and 18 August (b), over 2005-2020. While in (a) differences correspond to the QBO index for July defined at 10hPa, for (b) we chose the QBO index defined at 20hPa for August. Dots indicate significance at the 95% confidence level.**

So far, we have found a significant QBO signature on the temperature of the tropical Indian Ocean for the 16 Jun – 16 Jul time window, which is consistent with the AM water vapour response in July. However, in order to establish a causal link

between water vapour and temperature it is important to see whether they have a consistent temporal behaviour. Figure 2b represents the cross-correlation between the QBO index for July at 10hPa and AM water vapour (light blue line) and average Indian Ocean temperature (red line). Running windows of 31 days from 1 June to 30 September have been used to calculate the lagged correlations. These results evidence that while correlations reach a maximum centred in the month of July for water vapour (Fig. 2b, light blue line), the QBO impact on the tropical Indian temperature peaks around 15 days before (Fig.

2b, red line), which is consistent with the lag found between temperature and the interannual variability of the AM water vapour during July (Fig. 2b, dark blue line).

In contrast to what is observed for July, during August Figs. 1c-d reveals significant water vapour anomalies over the Asian Monsoon that are of opposite sign with respect to those over the equator. This is consistent with the QBO signal on the average temperature for the period 19 July-18 August shown in Fig. 3b, which depicts a branch of significant temperature

anomalies at subtropical latitudes with opposite sign compared to equatorial regions. These subtropical temperature anomalies extend over the southeastern edge of the AM anticyclone, from northern India to southwestern China, which is the region whose temperature, as shown in Fig. 2c, has a major impact on the interannual variability of the AM water vapour during August. The comparison of Figs. 1c-d and 3b evidence that temperature anomalies observed over this key region is in agreement with the QBO signature on water vapour over the AM. Thus, an increase in the AM water vapour of around 0.3

260    ppmv and 1 ppmv is accompanied by temperature anomalies reaching values between 0.5 K and 1 K over the region covering from northern India to southwestern China. Figures A2a and A3b in the appendix evidence the consistency between the QBO signal on the UTLS temperature over India and southwestern China and August water vapour in the AM for the QBO phases defined at different pressure levels. Fig. 2d allows us to analyze the temporal evolution of the impact of the QBO on water vapour in the AM in August and on the temperature over the region that we have identified as the one that

controls the interannual variability of water vapour over the AM, the southeastern edge of the AM anticyclone, from northern India to southwestern China (Fig. 2c). This figure represents the cross-correlation of the QBO index for August at 20hPa with the AM water vapour (Fig. 2d, light blue line) and the temperature over this precise region (Fig. 2d, red line), both averaged over 31-day running windows from June 1st to September 30th. The figure confirms that indeed the QBO signal on temperature peaks during the 19 July - 18 August time window (the 31-day window centered around the 3rd-4th of

August), with a lag of twelve days over the maximum in the QBO signal on the AM water vapour.

In order to explore further details of the three dimensional QBO temperature and wind patterns linked to the AM water vapour signature, Figs. 4a-b show the latitude-height cross sections of QBO-W minus QBO-E differences for temperature and zonal wind averaged over 60E-120E. After checking that ERA-5 reanalysis data reproduced the QBO signal on the temperature field at 100hPa (not shown), we used this dataset for the analysis of the three-dimensional temperature and wind

variations over the AM associated with the QBO signature on the AM water vapour. In order to obtain the QBO anomaly pattern with the greatest impact on AM water vapour, Fig. 4a shows QBO related differences for the QBO phases defined

according to the equatorial zonal wind of July at 10hPa and for the time window between 16 Jun and 16 Jul while, for August Fig. 4b depicts differences for the QBO phases defined according to the August wind at 20hPa and for the time window between 19 Jul and 18 Aug. In agreement with the chosen QBO phase definition, Figs. 4a-b show westerly wind anomalies centered at 10hPa and 20hPa and easterly anomalies centered at 50 and 70hPa respectively. The QBO also exhibits a signature in temperature in both tropics and subtropics. The tropical QBO temperature is in thermal wind balance with the vertical shear of the zonal winds and, according to this, Figs. 4a and 4b show cold temperature anomalies in regions of the tropical UTLS that exhibit easterly wind shear. As it has been established in previous studies (Baldwin et al., 2001), besides the equatorial maximum in QBO temperature, out of phase anomalies may appear over 20º-40ºN associated with the secondary meridional circulation. Despite the fact that these subtropical anomalies are weaker in the summer hemisphere, the global zonal average for 16 June - 16 July shows warm anomalies in the region of the UTLS at these latitudes (Fig. 4c). However, these anomalies are not found for the zonal average between 60E-120E (Fig. 4a), covering the southern edge of the AM anticyclone, where only weak cold anomalies are found at 100hPa surrounded at upper and lower levels by weak warm anomalies that, in either case, are not statistically significant. On the contrary, during August, warm anomalies in the region of the UTLS over the southern flank of the monsoon (Fig. 4b) are stronger than those found for the global average (Fig. 4d). Thus, while figure 4b depicts a warming between 1 and 1.5ºC at 100hPa and over 20N-40N, in agreement with temperature anomalies in figure 3b, global zonal mean anomalies are below 0.75ºC in this region (Fig. 4d). Furthermore, Fig. 4b evidences that subtropical temperature anomalies over the longitude range corresponding to the AM extend over 20N-40N and from 100hPa to 50hPa and reveals that they are part of a set of temperature anomalies that form a wave train-like pattern that extends from the tropics to high latitudes of the northern hemisphere in thermal wind balance with zonal wind anomalies. Over the AM anticyclone latitude range, from 10N to 50N (Fig. 4e), these temperature anomalies are characterized, in the lower stratosphere, by positive and negative anomalies in the southern and northern flanks of the AM anticyclone and out of phase anomalies in the troposphere.  Fig. 4b also shows consistent variations of the zonal wind in thermal wind balance with the temperatures. When comparing these zonal wind anomalies with the climatological mean (Fig. 4e), it is clear that wet anomalies (as evident from Figs. 1c-d) are linked to a weaker anticyclone, which in turn is linked to a cold troposphere and warm lower stratosphere in the latitude range between 15-35ºN and temperature anomalies of opposite sign north of 35ºN (Fig. 4b). This is consistent with Randel et al. (2015), who found that the intraseasonal variability of the AM water vapour was linked to a similar pattern of temperature and zonal wind anomalies.

So far, our results demonstrate that the QBO modulation of the lower stratosphere temperatures over certain key regions precede consistent water vapour variations over the AM. These temperature variations, in turn, provoke a QBO signature on the AM water vapour through large scale dehydration that is in phase with the signature over the equatorial and tropical region during July and out of phase during August. The reason for this intra-seasonal change in the AM water vapour signal is explored in the next section.

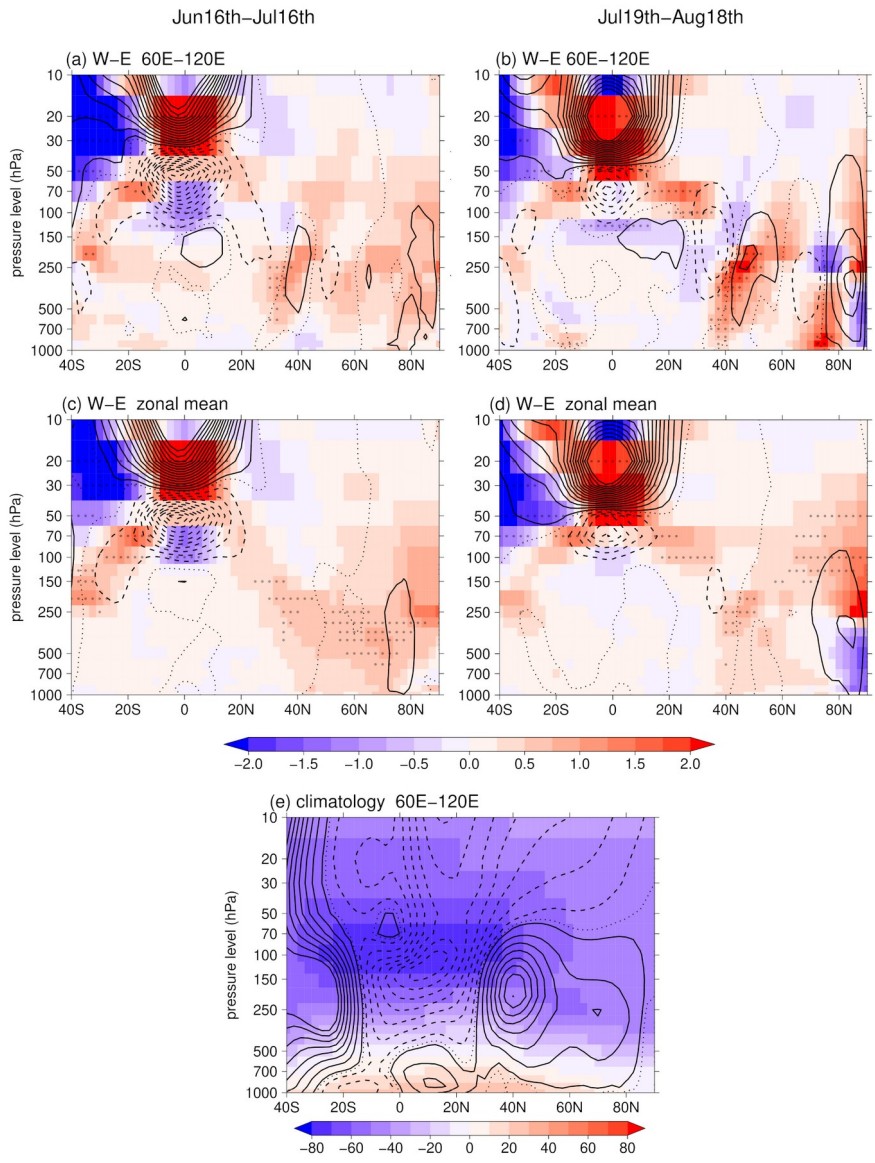

**Figure 4: Latitude-height cross sections of QBO-W minus QBO-E differences for ERA5 temperature (colors) and zonal wind (black contours) averaged over 60E-120E and between June 16th and July 16th (a) and between July 19th and August 18th (b). In (a) diifferences correspond to the QBO index for July defined at 10hPa while for (b) we chose the QBO index defined at 20hPa for**
**August. Solid/dashed contour lines show positive/negative anomalies with contour intervals at every 2m/s from 1m/s/-1m/s for positive/negative anomalies. Figures (c) and (d) are equivalent to (a) and (b) but for the zonal mean temperature and zonal wind. Dots indicate significance at the 95% confidence level. Figure (e) the climatology for the zonal wind and temperature averaged over 60E-120E for the boreal summer (June, July and August). Solid/dashed contour lines show positive/negative values with contour intervals at every 4m/s from 1m/s/-1m/s for positive/negative anomalies.**

## 4 The role of the QBO impact on clouds

Previous studies have found that the QBO modulates clouds and convection over the AM (Giorgetta et al., 1999; Peña-Ortiz et al., 2019; Sweeney et al., 2023). In addition, Randel et al. (2015) showed evidence that temperature variations that precede the intraseasonal changes in AM water vapour are linked to convection changes. In this way, they found that enhanced convection over the southeastern AM produced a colder UTLS over this region, giving rise to drier conditions over the AM. These results raise the question of whether a similar mechanism operates on an interannual scale and, more specifically, whether convection plays a role in the transmission of the QBO signal to AM water vapour. To address this question, we have made use of OLR and fraction of cloud cover, in order to characterize the QBO signature on clouds, and also of diabatic heating rates, to determine the relation between changes in clouds and its possible impact on circulation and temperature. With this purpose, we have computed QBO-W minus QBO-E differences for the same time windows as used for the temperature, between 16 Jun and 16 Jul for the analysis of the QBO signal on water vapour in July and for the average between 19 Jul and 18 Aug for August.

Figure 5a shows QBO-W minus QBO-E (defined at 10hPa) differences for the fraction of cloud cover averaged between 16 Jun and 16 Jul and between 100hPa and 150hPa. This range of levels were chosen after verifying that the QBO signal on cloudiness significantly weakens below 150hPa (Fig. 5b), which is consistent with previous studies (Giorgetta et al., 1999; Peña-Ortiz et al., 2019). Figure 5a shows positive anomalies of fraction of cloud cover between 100hPa and 150hPa over the equatorial Indian Ocean and Indonesia that might indicate a convection increase during the westerly phase of the QBO (as defined at 10hPa, corresponding to an easterly phase around 100-70hPa, see Fig. 4) that can reach about 10% over some areas. In agreement with these results, negative OLR anomalies are found over this region during the QBO-W compared to QBO-E (Fig. A4a). These are consistent with previous studies (Giorgetta et al., 1999; Collimore et al., 2003; Peña-Ortiz et al., 2019) evidencing that the temperature anomaly initiated by the adiabatic temperature change due to the secondary circulation of the QBO can modulate deep convection in a way that the UTLS cooling linked to the easterly QBO jet at lower levels around 100-70hPa causes a lower static stability that allows deep convection to develop more vigorously. Although these studies attributed the QBO signal on cloudiness in the upper troposphere to changes in deep convection, our results do not allow us to determine on which type of clouds the QBO is acting and whether the observed signal corresponds to changes in convection or in the occurrence of cirrus clouds. Sweeney et al. (2023) showed that the QBO primarily affects cloudiness above 200hPa, mainly impacting cirrus clouds. However, their study also revealed a QBO signal on the upper troposphere cloudiness associated with opaque clouds, which are often linked to deep convection and thick anvil. Accordingly, Giorgetta et al. (1999) argued that QBO acts primarily by raising the height of cloud tops, which, when the QBO causes a cooling of the tropopause, can more frequently reach levels between 100hPa and 150hPa rather than between 150hPa and 200hPa. Thus, it is highly likely that QBO related differences on fraction of cloud cover and on OLR obtained in the present study reflect both the QBO signal over cirrus clouds and also an intensification of convection in the upper levels of the troposphere.

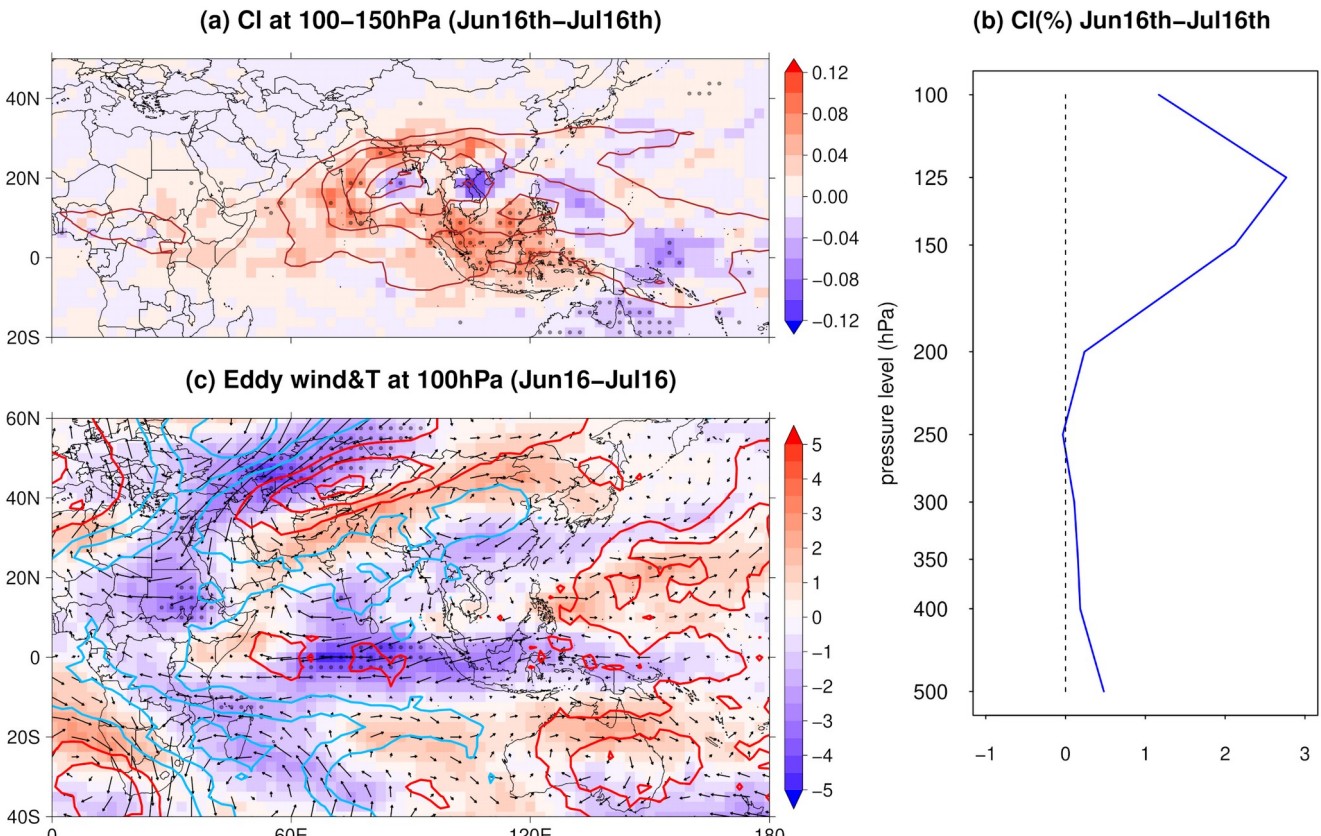

Figure 5: (a) QBO-W minus QBO-E differences for the average over the period 16 Jun – 16 Jul of fraction of cloud cover averaged between 100hPa and 150hPa in parts per unit. Red solid lines represent the climatological average for the same time window and over the period 2005-2020 with contour intervals at every 0.1 from 0.1 in parts per unit. (b) Similar differences but for fraction of cloud cover averaged over 60E-120E and 15S-25N, the region with the largest anomalies, at different pressure levels from 500hPa to 100hPa. (c) Equivalent differences for eddy fields of zonal wind (color shades) and horizontal wind (arrows). Blue/red contour lines show negative/positive anomalies of the temperature eddy field with contour intervals at every 0.5K starting at 0.25K and -0.25K for positive and negative anomalies respectively. In (a) and (c), dots indicate significance at the 95% confidence level.

Figures. 5a and A4a reveal that the impact of the QBO on cloud cover is not limited to the equatorial Indian Ocean, where temperature variations associated with the secondary circulation of the QBO can explain this signal. Beyond the equator, a significant increase in cloud cover during the westerly phase of the QBO (defined at 10hPa) is also observed over most of India (Fig. 5a). These cloud cover changes show increases around the core region of climatological convection and decreases within the core region, indicating an increase in convective area. The increase in cloud cover is associated with an intensification of the anticyclonic circulation over this area, as seen in Fig. 5c showing the 100hPa zonal wind eddy field.

Our results suggest that the anticyclonic anomaly and the convection increase over India can be part of the response to the QBO modulation of the equatorial clouds. The possibility that changes in the upper troposphere clouds may have an impact on circulation and on the excitation of wave trains, was addressed in Slingo and Slingo (1991), Giorgetta et al. (1999) and Peña-Ortiz et al. (2019) among others. These studies show that diabatic heating caused by cloudiness at the upper troposphere, including that associated with tropical cirrus clouds, can excite wave trains that can propagate to higher latitudes. Slingo and Slingo (1991) found that the major dynamical response to upper tropospheric diabatic heating is restricted to the tropics and the subtropical jets and that this response is characterized by an anticyclonic dipole to the north and south of the diabatic heating that can excite wave trains propagating to higher latitudes. This response is possibly associated with the excitation of an internal Rossby mode as previously described by Matsuno (1966) and Gill (1980).

To verify whether a similar mechanism is at work here and whether the observed increases in cloudiness generate latent heat release, we computed QBO related differences for diabatic heating rate averaged between 16 Jun and 16 Jul and for two different layers: between 100hPa and 700hPa and for 100hPa separately. The pattern of diabatic heating rate anomalies resembles that of cloud cover fraction (Figs. 5a and 6a), with positive anomalies indicating latent heat release over areas of increased cloud cover such as the northern Indian Ocean and most of India during QBO-W in comparison with QBO-E. Figure 6b shows positive diabatic heating rate anomalies at 100hPa, that are most intense and statistically significant to the south of the Bay of Bengal. The comparison of Figs. 6b and 5c reveals that these positive diabatic heating rate anomalies are associated with the intensification of easterly winds just west of the area of latent heat release. These easterly wind anomalies over the Indian Ocean, just to the south of India, together with the westerly wind anomalies to the north of this region, are part of an anticyclonic gyre which contributes to the strengthening of the climatological AM anticyclonic circulation giving rise to an enhancement of the rising motions and adiabatic cooling, as can be inferred from the cold anomalies associated with the anticyclonic gyre (Fig. 5c). Although westerly wind anomalies are also observed in the Southern Hemisphere just south of the equatorial easterly wind anomalies, the southern counterpart of the anticyclonic dipole described by Slingo and Slingo (1991) is not clearly observed in Fig. 5b, which could be due to the stronger latent heat released over the northern tropical Indian Ocean (Fig. 6b).

**(a) dhr at 100–700hPa (Jun16th–Jul16th)**

**(b) dhr at 100hPa (Jun16th–Jul16th)**

**Figure 6: QBO-W minus QBO-E differences for diabatic heating rate in K/day (color shades) averaged between 100hPa and 700hPa (a) and at 100hPa (b) for the period 16th June - 16th July. Blue/red contour lines show negative/positive anomalies of the temperature eddy field with contour intervals at every 1K/day starting at 1K/day and -1K/day for positive and negative anomalies respectively. Dots indicate significance at the 95% confidence level.**

Eddy temperature anomalies observed in Fig. 5c show cooling anomalies over an extended region covering the southern edge of the AM, from India to the north of the Bay of Bengal, linked to the anticyclonic anomaly, of up to 1 K, which is slightly higher than the cooling diagnosed from the total temperature field in Fig. 3 (wavefield plus zonal average). The reason for this difference is the zonal mean temperature, which shows a warming associated with the secondary circulation of the QBO over subtropical latitudes (Fig. 4a) and partially compensates eddy temperature anomalies. As a result, a slight
(and not statistically) temperature decrease is found over the southern edge of the AM (Fig. 3). The fact that these temperature anomalies are weak might explain why the temperature of this region, which previous studies point to as the main cause of dehydration of the air masses reaching the AM (Wright et al., 2011; Randel et al., 2015), is not that relevant for AM water vapor changes during July, while the temperature over the Indian Ocean is (Fig. 2a). By contrast, this weak subtropical cooling occurs simultaneously with a strong tropical temperature decrease associated with the secondary
circulation of the QBO (Fig. 3), such that, the entire region whose temperature modulates the transport of water vapour in the AM according to Fig. 2a, shows a cooling during the westerly phase of the QBO (as determined by the 10 hPa tropical zonal wind). This explains why the signal on water vapour in the AM has the same sign and characteristics as the signal on

equatorial water vapour and why in both regions a decrease in water vapour concentration is observed during QBO-W in relation to QBO-E.

Concerning the QBO modulation of the AM water vapor during August, QBO-W minus QBO-E (defined at 20hPa) differences for the average between 19 Jul and 18 Aug of cloud area fraction (Figs. 7a) reveal quite a different behavior with respect to the anomalies for 16 June – 16 July (Figs. 5a). A northward and westward shift of the anomalies is observed during 19 Jul – 18 Aug, suggesting that the impact of the QBO on clouds weakens over the equator and intensifies at latitudes higher than 15N as the boreal summer progresses. Figure 7a evidences a dipole structure characterized by a

decrease/increase of cloud cover over the southeastern/southwestern flank of the AM (centered around 25ºN and between 60ºE–100°E) during QBO-W compared to QBO-E, which is consistent with the patterns found for OLR and diabatic heating rate anomalies (Figs. A4b and A5). Figure 7b shows that, as for June/July (Fig. 6d), cloud area fraction anomalies are restricted to the atmospheric layer between 200-100hPa. The cloud cover decrease observed in Fig. 7a is linked to a cyclonic anomaly (Fig. 7c) and occurs over the area of the climatological maximum in August, north of the Bay of Bengal. To the

west of this area, positive anomalies of cloud area fraction indicate an increase of cloud cover, suggesting a westward shift of the convective activity during QBO-W in comparison with QBO-E.

Figure 7c reveals that the cloud cover decrease observed to the north of the Bay of Bengal (Fig.7a) is linked to a Rossby wave train that generates the cyclonic gyre over this region. The QBO signature on the eddy zonal wind field reveals a Rossby wave train that propagates from the equator to extratropical latitudes over both hemispheres between the eastern

Indian and the western Pacific oceans. This figure, which also shows QBO related differences in the eddy temperature field at 100hPa, evidences that the decrease of cloud cover to the north of the Bay of Bengal, the region of the climatological maximum, appears together with a warming over the southern edge of the AM. This warming occurs over the region that, as previously shown in figure 2c, controls the inflow of water vapor into the AM. On the other hand, QBO-W minus QBO-E differences of the global zonal mean temperature averaged over the period 19 Jul and 18 Aug (Fig. 4d) show a warming in

the lower stratosphere between 15N and 35N, corresponding to the subtropical branch of the secondary circulation of the QBO. Thus, during this time window, temperatures over the southern flank of the AM show an increase during the QBO westerly phase defined at 20hPa (Fig. 3b), which is observed in both the eddy field and the zonal mean. This suggests that the warming is caused by both a Rossby wave train associated with the QBO that weakens rising motions over this region and also by the secondary meridional circulation of the QBO. Therefore, both mechanisms contribute to the warming that

gives rise to the water vapour increase found during August over the AM.


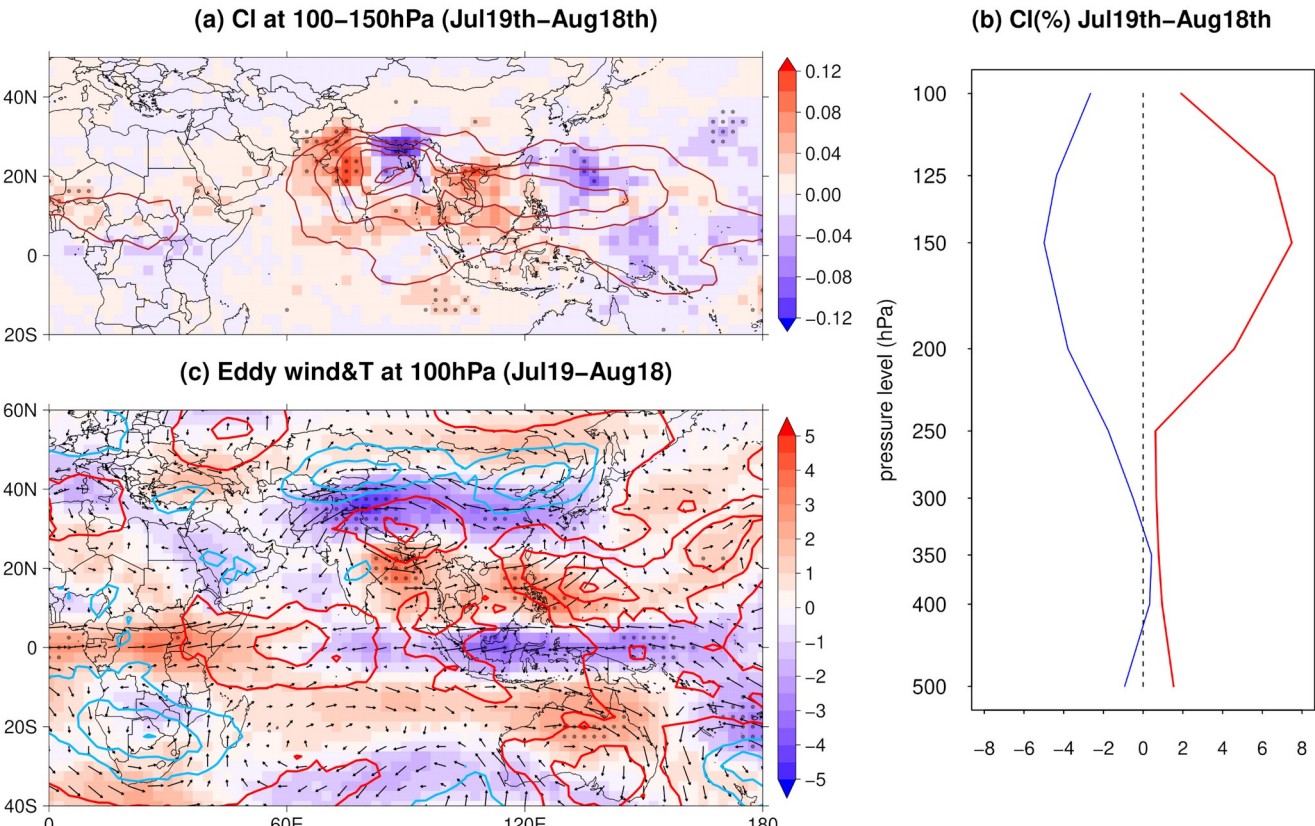

**Figure 7: (a) QBO-W minus QBO-E differences for the average over the period 19 Jul – 18 Aug of fraction of cloud cover averaged between 100hPa and 150hPa in parts per unit. Red solid lines represent the climatological average for the same time window and over the period 2005-2020 with contour intervals at every 0.1 from 0.1 in parts per unit. (b) Similar differences but for fraction of cloud cover averaged over 60E-80E and 20N-30N (red line) and over 80E-95E and 20N-30N (blue line), regions with the largest anomalies, at different pressure levels from 500hPa to 100hPa. (c) Equivalent differences for eddy fields of zonal wind (color shades) and horizontal wind (arrows). Blue/red contour lines show negative/positive anomalies of the temperature eddy field with contour intervals at every 0.5K starting at 0.25K and -0.25K for positive and negative anomalies respectively. In (a) and (c), dots indicate significance at the 95% confidence level.**

Giorgetta et al. (1999) found that the QBO modulation of deep convection over the tropical eastern Indian Ocean and western Pacific gives rise to the excitation of a wave train. They observed that despite the fact that this wave train reached its mature phase in July/August, it was forced by equatorial diabatic heat release in the previous weeks by the QBO modulation of equatorial deep convection. In order to study the possible link between the wave train observed in Fig. 7c and a hypothetical QBO signature on equatorial clouds in the previous weeks, we computed QBO-W minus QBO-E composites for diabatic heating rate and the eddy field of horizontal wind at 100hPa for different 31-day windows prior to the one

between 19 Jul - 18 Aug, using always the QBO index defined for August at 20hPa. In this way, we can assess the possible impact of the QBO on the circulation through cloud modulation in the weeks preceding the formation of the wave train

observed in Fig. 7c. Additionally, we also computed similar differences for the total and eddy temperature fields at 100hPa in order to see the evolution of the signal over the summer and its dependence on the circulation. Figure 8 displays the obtained results for the time windows between 19 Jun – 19 Jul, 29 Jun – 29 Jul, 9 Jul – 8 Aug and 19 Jul - 18 Aug. This figure reveals that in the weeks preceding the period 19 Jul-18 Aug, strong positive diabatic heating rate anomalies occur during QBO-W compared to QBO-E over the eastern equatorial Indian Ocean extending over the Bay of Bengal and

southern India, indicating the release of latent heat associated with increased cloudiness. This diabatic warming causes strong easterly wind anomalies to the west of the region of latent heat release, which are part of an anticyclonic gyre extending over China and India. In addition, positive and negative zonal wind anomalies alternate north and south of this region of latent heat release forming a wave train that reaches high latitudes in both hemispheres in the longitudinal sector between 40E and 100E. In relation to temperature, we find a cooling over the same region where the latent heat release is

taking place, which is explained by adiabatic upwelling and longwave cooling associated with the cloudiness increase. Also negative temperature anomalies are found over India and western China, linked to the anticyclone structure found over this region, ranging between -0.5 ºC and -1 ºC for the eddy temperature field and keeping above -0.5 ºC for the total temperature. As the summer progresses, the impact of the QBO on cloudiness over the eastern equatorial Indian Ocean weakens, as seen in the progressive weakening of the positive diabatic heating rate anomalies in Figs. 8a-d. At the same time easterly wind

anomalies over the Indian Ocean and India also weaken and become restricted to the equatorial region (Figs. 8c-d). The wave train shown in Figs. 8a-b is also found in Figs. 8c-d but the alternating easterly and westerly zonal wind bands in the northern subtropics seem to shift southwards as the equatorial easterlies weaken. In this way, the anticyclonic structure previously observed over India and western China (Fig. 8 a-b), progressively turns into a cyclonic gyre (Figs. 8 c-d). With regard to temperature, while Figs. 8 e-f showed a cooling associated with the anticyclonic anomaly, Figs. 8g-h show the

appearance of a warm anomaly centered north of the Bay of Bengal associated with the cyclonic gyre. This is in line with the fact that the cyclonic anomaly implies a weakening of the climatological anticyclone over this area of the AM and, therefore, a weakening of the rising motions as well as of the adiabatic cooling.

It should be noted that we have addressed the role of the QBO modulation of clouds through their impact on circulation and temperature. However, the observed changes in cloud cover may be linked to deep convection and can also have an impact

on the AM water vapour via direct overshooting. This pathway has not been addressed in the present study but may be a fruitful topic for future research.

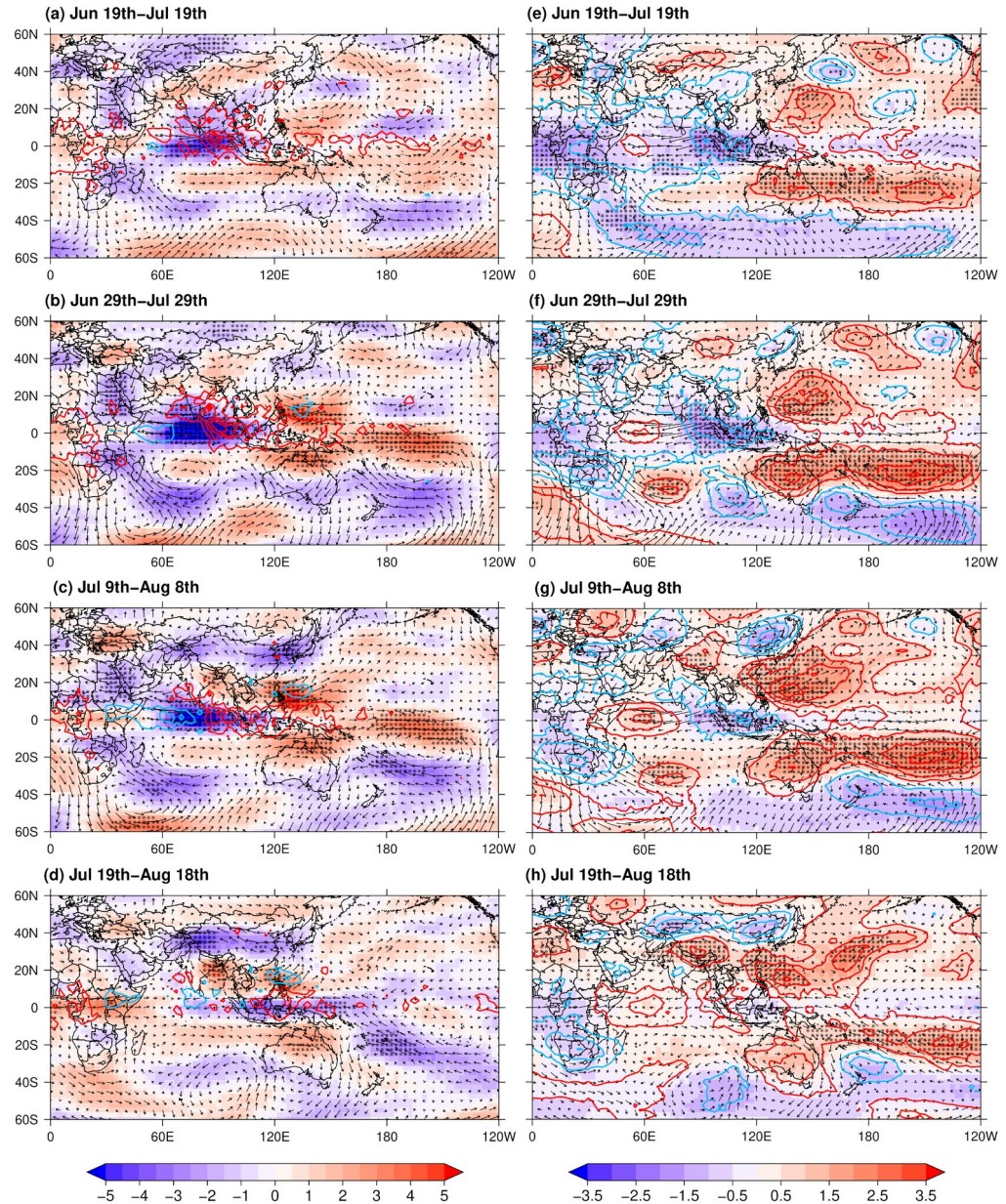

**Figure 8: QBO-W minus QBO-E differences for diabatic heating rate in K/day (contours) and for the eddy fields of zonal wind in m/s (color shades) and horizontal wind (arrows) at 100hPa averaged over the period between (a) 19th June and 19th July, (b) 29th June and 29th July, (c) 9th July and 8th August and (d) 19th July and 18th August. Blue/red contour lines show negative/positive anomalies with contour intervals at every 0.5K/day starting at 0.25K/day and -0.25K/day for positive and negative anomalies respectively. Figures (e)-(h) are equivalent to (a)-(d) but for the temperature (color shades) and the eddy fiels of temperature in**
**K(contours). Blue/red contour lines show negative/positive anomalies with contour intervals at every 0.5K starting at 0.25K and -0.25K for positive and negative anomalies respectively. Dots indicate significance at the 95% confidence level.**

**5 Summary and conclusions**

The Asian Monsoon plays a key role in the transport of water vapour to the lower stratosphere and contributes significantly to the wet phase of the annual global stratospheric water vapour cycle. The interannual variability of the lower stratospheric water vapour over the Asian Monsoon is dominated by the QBO and ENSO (Randel et al., 2015). However, the physical mechanisms responsible for this variability have been poorly investigated.

Here we have made use of daily MLS data for the period 2005-2020 to characterize the QBO signature on the lower stratosphere AM water vapour during the boreal summer. We have found that the QBO has the strongest impact during August, although a significant signature is also observed during July. In July, the QBO modulation of the AM water vapour occurs in phase with the modulation of the lower stratospheric water vapour over the equator. Hence, as the equatorial UTLS cools as a response to the QBO winds, the water vapour over the tropical lower stratosphere and the AM decreases via large scale dehydration and vice-versa. This synchrony is related to the fact that, in this month, the region whose temperature controls the water vapour variability associated with the QBO over the AM is the UTLS over the tropical Indian Ocean, which also modulates the inflow of water vapour across the equator over this region. Although the south side of the AM anticyclone is a key region in controlling water vapour over the Asian monsoon (Wright et al., 2011; Randel et al., 2015), we find that in July the QBO signal on temperature is weak over this region and only significant over western India and the Arabian Sea, which shows anomalies of the same sign as the signal over the equator. These results suggest that the AM water vapour signal in July responds to the QBO temperature over a large region stretching from the tropical Indian Ocean to the northern Arabian Sea and western India, where the QBO signal on temperature is in phase. Furthermore, our results show a time lag between the temperature and water vapour signal over the AM of about 15 days. In this way, temperature anomalies over the tropical Indian Ocean of -1 K and -2 K precede a water vapour decrease of around -0.4 ppmv that can reach -0.8 ppmv to the west of India.

Conversely, in August the region whose temperature controls water vapour inflow to the AM is primarily the southern edge of the AM anticyclone, from northwestern India to southeastern China. This is consistent with Randel et al. (2015), who pointed to the temperature over this region as the main driver of the intraseasonal variability of AM water vapour. Moreover, in contrast with the QBO signal on temperature in July, during August, the UTLS over this region shows temperature anomalies associated with the QBO of opposite sign to those over the equator and, for this reason, the QBO signal on the lower stratospheric water vapour over the equator shows an opposite sign to the signal over the AM. The signal in the temperature of this region, the southern edge of the AM anticyclone, shows a warming during the QBO westerly phase compared to the easterly phase of 0.5 K to 1 K, which, with a time lag of about 12 days, leads to relatively less dehydration and water vapor increase over the monsoon reaching values between 0.3 and 1 ppmv.

Regarding the mechanism involved in the observed patterns, our results suggest that the QBO impact on the temperature at the southern flank of the AM and, consequently, on the AM water vapour during July and August, is modulated by the QBO impact on clouds. For the QBO signature on the AM water vapour during July, our results show that, during the preceding

weeks, QBO cold anomalies over the equator cause an increase in cloud cover at levels between 150hPa and 100hPa leading to latent heat release over the eastern Indian Ocean. Eddy temperature and wind fields show a response to these tropical anomalies of latent heat that produce a Rossby anticyclonic gyre at 100hPa over India, which intensifies rising air motions and gives rise to a cooling over this region. This cooling, which is clearly observed in the eddy temperature field, is partially balanced by the zonal mean temperature, which shows a warming associated with the secondary circulation of the QBO over

subtropical latitudes. As a result, the total temperature field over the southern flank of the AM shows a slight and not significant cooling during QBO-W (defined at 10hPa). This absence of a strong QBO impact on subtropical temperatures over the southern flank of the AM, explains the reason why during July the QBO signature on the AM water vapour is modulated by QBO temperature anomalies over the equator and the fact that the QBO modulation of AM water vapour is in phase with its modulation of the lower stratosphere water vapour over the equatorial region.

Our results also evidence a relationship between the QBO modulation of the AM water vapour during August and the QBO modulation of convection. Temperature anomalies over the southern flank of the AM, preceding the QBO-associated changes in AM water vapour, are consistent with the observed changes in cloudiness. In this way, the observed warming over this region, that controls the inflow of water vapour into the monsoon, is accompanied by a decrease in cloud cover in the 100-150hPa layer over to the north of the Bay of Bengal. At the same time, an increase in cloud cover is observed to the

west of this region, over northwest India. Although our analysis does not allow us to determine whether the cloud cover anomalies correspond to clouds of convective origin, the fact that the reduction in cloudiness occurs north of the Bay of Bengal, the region of the climatological maximum of convection, and that it appears associated with a cyclonic anomaly suggest that water vapour changes over the Asian Monsoon are associated with a westward shift of convection, characterized by anomalously strong convection in the southwest and weak convection in the southeast of the AM. Remarkably, previous

studies (Randel et al., 2015; Zhang et al., 2016) found a dipole pattern in intraseasonal convective variability very similar to the pattern found here for QBO-related variability, with similar effects on the AM water vapour. In this regard, Randel et al. (2015) pointed out the apparent contradiction arising from the fact that a reduction of convection over the region of the climatological maximum is associated with an increase, rather than a reduction, of humidity over the AM. Randel et al. (2015) demonstrate that the intra-seasonal water vapour variability in the AM is related to the upper-level temperature

response to convective variability in that region (i.e. cooling/warming due to enhanced/weakened convection), which is key to the dehydration of the air parcels reaching the AM. Our results show a similar relation between the QBO response in AM water vapor and convection, particularly a warm anomaly over the region where the reduction in cloud cover appears. Furthermore, Zhang et al. (2016) found that, at intraseasonal time scales, the intensification of convection over the western monsoon edge implies an increase in upward motions over this region, which is warmer than the southeast flank of the AM.

They showed that this westward shift of the AM convective systems favours the entry of air masses into the AM through this region, allowing them to transport a higher water vapour amount. Our results suggest that a similar mechanism also contributes to the QBO impact on the AM water vapour during August.

Our results show that the modulation of cloud cover over the southeastern flank of the AM during August, is linked to a Rossby wave train associated with the QBO that propagates from the equator to extratropical latitudes over both hemispheres

between the eastern Indian and the western Pacific oceans. Thus, the cloud cover decrease found over the region of the climatological maximum when QBO westerlies dominate at 20hPa, is linked to a cyclonic gyre over this region that forms part of the wave train, as it can be observed in the eddy wind field at 100hPa. Furthermore, in agreement with Giorgetta et al. (1999), our results suggest that this wave train emerges in response to the QBO modulation of convection over the tropical Indian Ocean in the preceding weeks.

The observed differences in the impact of the QBO on AM water vapour in July and August are consistent with the changes in the QBO signal on equatorial clouds in these two months. In accordance with previous studies, we have observed that it is temperature over the southern flank of the AM that has the greatest impact on the moisture over the AM anticyclone. In turn, the QBO signal on temperature in this region is linked to the modulation of equatorial clouds. However, while in the period from mid-June to mid-July the temperature signal over the southern flank of the AM occurs in response to the signal on

clouds occurring simultaneously over the equator, from mid-July to August the temperature signal over the southern flank of the AM seems to be associated with a Rossby wave train generated by the QBO modulation of equatorial clouds in the previous weeks. The fact that the modulation of cloud cover by the QBO occurs mainly over the region of the climatological maximum of convection and that only affects at levels above 200hPa are in line with the results of Giorgetta et al. (1999) showing that the QBO cooling of the equatorial UTLS causes a lower static stability that allows deep convection to develop

more vigorously reaching up to 150 hPa or higher. In accordance with this premise, the QBO would need the presence of significant convective activity to have an impact on the vertical extent of convection modulating atmospheric stability. Thus, as the climatological maximum of convection moves northward with the advance of summer and weakens at the equator (Fig. 9), the impact of the QBO on convection also weakens at equatorial latitudes. Therefore, for the period considered, these results suggest that the temporal evolution over the summer of the impact of the QBO on clouds over the eastern

equatorial Indian Ocean has an impact on the circulation and temperature on the southern flank of the AM. Thus, when in early summer, the cooling of the tropical tropopause associated with the QBO favors an increase in cloud cover and a release of latent heat, a wave train is formed, characterized by an anticyclonic anomaly centred over the northern Bay of Bengal and a cold anomaly in the temperature over the southern flank of the AM that produces a dry anomaly over the AM. The progressive weakening of the QBO signal on equatorial clouds causes a weakening of the easterly wind anomalies associated

with the wave train over the tropical region so that, during the time window covering the second half of July and the first half of August, the disturbance over the southern flank of the AM associated with this wave train is characterized by a cyclonic gyre that inhibits convection and causes a warm anomaly that moistens the AM.

Remarkaby, the general response of water vapour in the AM to variations in convection on both intra-seasonal and inter-annual time scales is very similar. Hence, the involved mechanisms, as detailed in this paper, could also help to explain

changes in stratospheric water vapour in the AM region for convection changes in a changing climate. Follow-up studies on the climate change response of monsoon moisture will be a fruitful topic of future research.

It should be noted that we have assessed the QBO signal on water vapour as a consequence of its impact on temperature around the tropopause and the local dehydration it can cause. However other processes, such as changes in transport could also be involved. A detailed analysis of the possible changes caused by the QBO on the trajectories of the air masses reaching the AM, or the variations of such trajectories with the summer progress, could also explain part of the observed QBO signature on the AM water vapour or the intraseasonal evolution of this signature. Future studies incorporating Lagrangian transport models to address this question may determine the role of this pathway. Another limitation of the present study derives from the scarcity of satellite data of water vapour in the lower stratosphere, with series starting only in 2005. The number of QBO-W and QBO-E cases obtained for our period of study has made it possible to identify a significant QBO signal on the AM water vapour induced by the QBO modulation of the lower stratosphere temperature. However, future research with longer data series is required for a more robust assessment of the intraseasonal variability of this signature and its relation with the QBO impact on tropical clouds, which may be subject to interaction with other patterns of variability such as ENSO or the Madden-Julian oscillation.

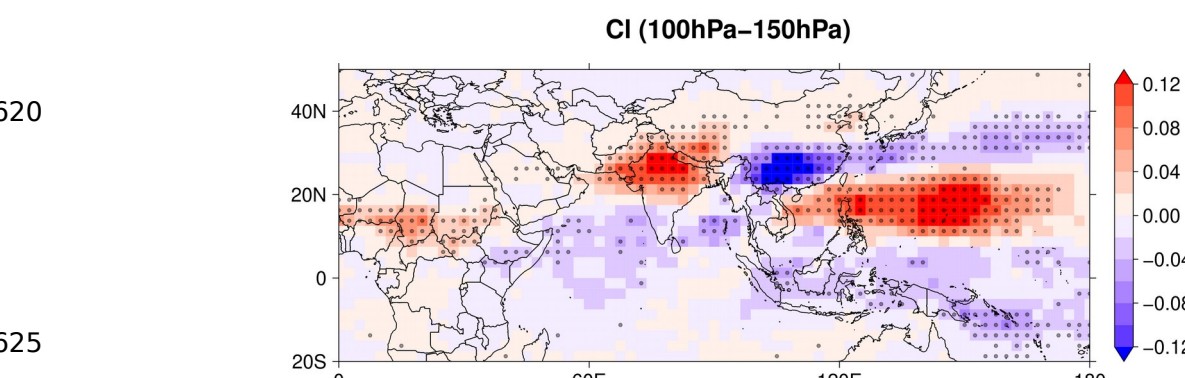

**Figure 9: Difference between the climatological mean over the period 2005-2020 of the mean cloudiness between 100hPa and 150hPa corresponding to the time interval between 16 June and 16 July minus the average corresponding to the period 19 July - 18 August.**

## Appendix A

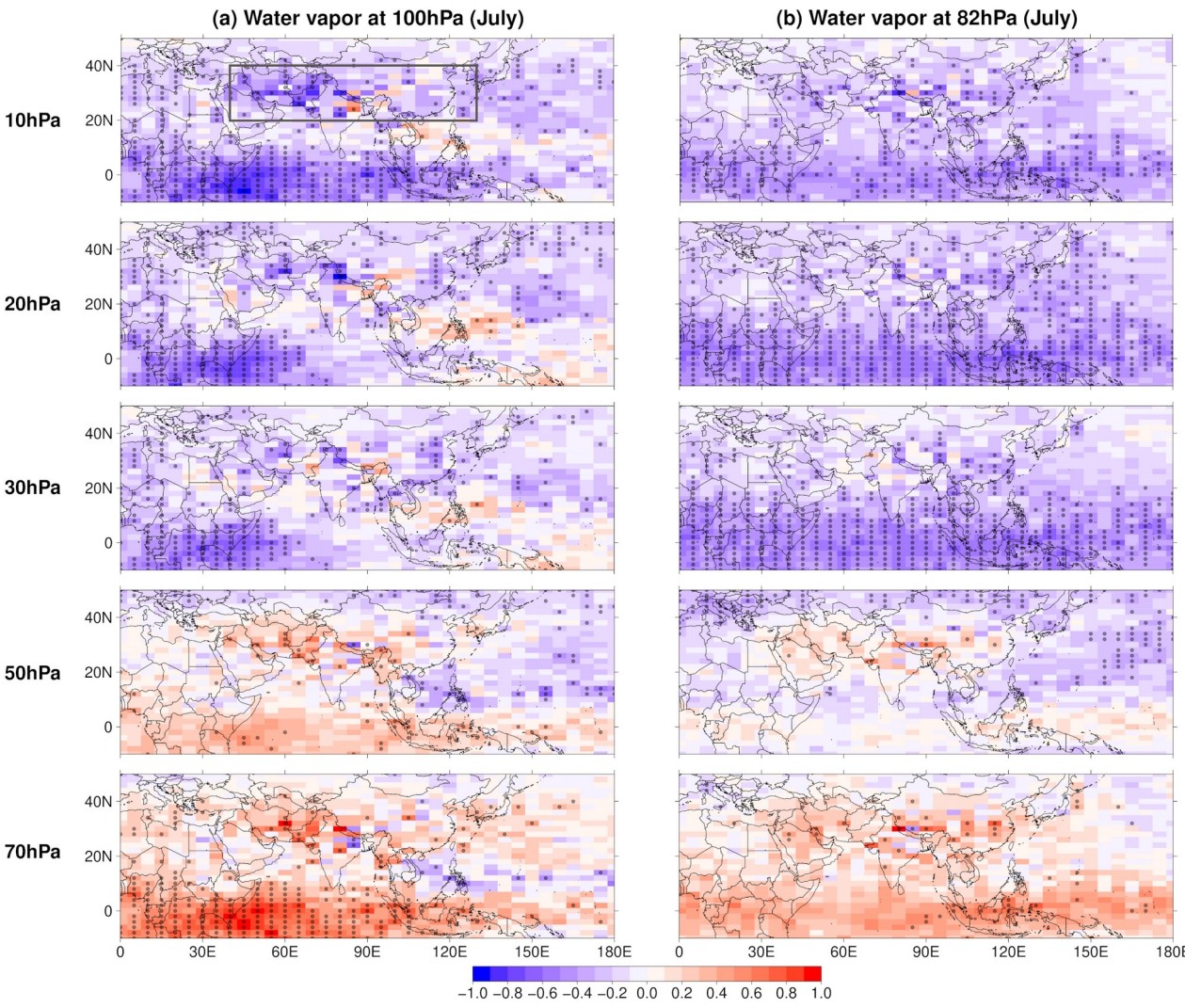

**Figure A1: QBO-W minus QBO-E differences for MLS water vapour at 100hPa (a) and 82hPa (b) for July over the period 2005-2020. Each row corresponds to a different level at which the QBO phases were defined from 10hPa (top row) to 70hPa (bottom row). Dots indicate significance at the 95% confidence level. The region inside the grey box corresponds to the Asian monsoon region during the QBO phase in which we have identified a stronger signal on the water vapour of this area.**

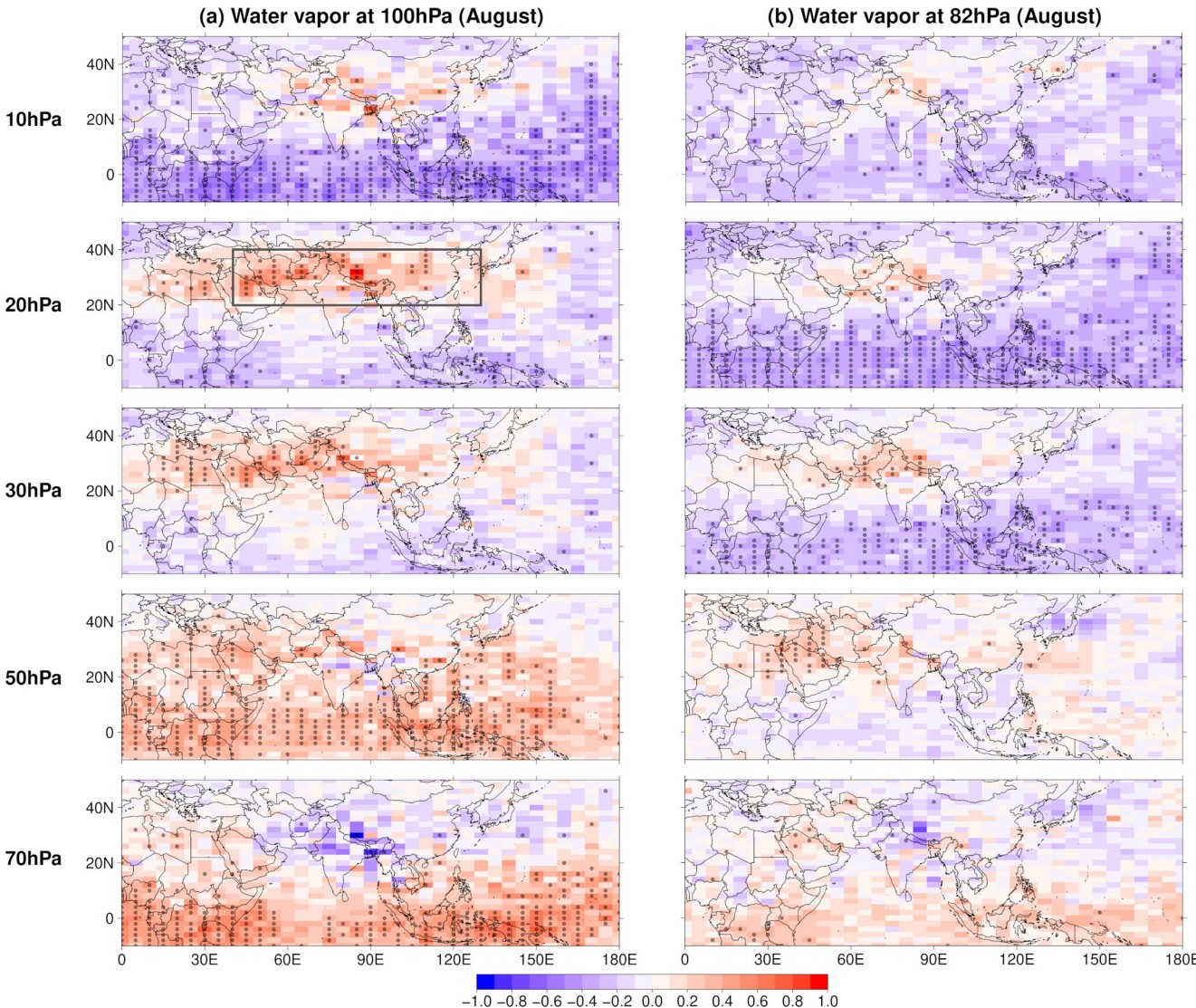

**Figure A2: QBO-W minus QBO-E differences for MLS water vapour at 100hPa (a) and 82hPa (b) for August over the period 2005-2020. Each row corresponds to a different level at which the QBO phases were defined from 10hPa (top row) to 70hPa (bottom row). Dots indicate significance at the 95% confidence level. The region inside the grey box corresponds to the Asian monsoon region during the QBO phase in which we have identified a stronger signal on the water vapour of this area.**

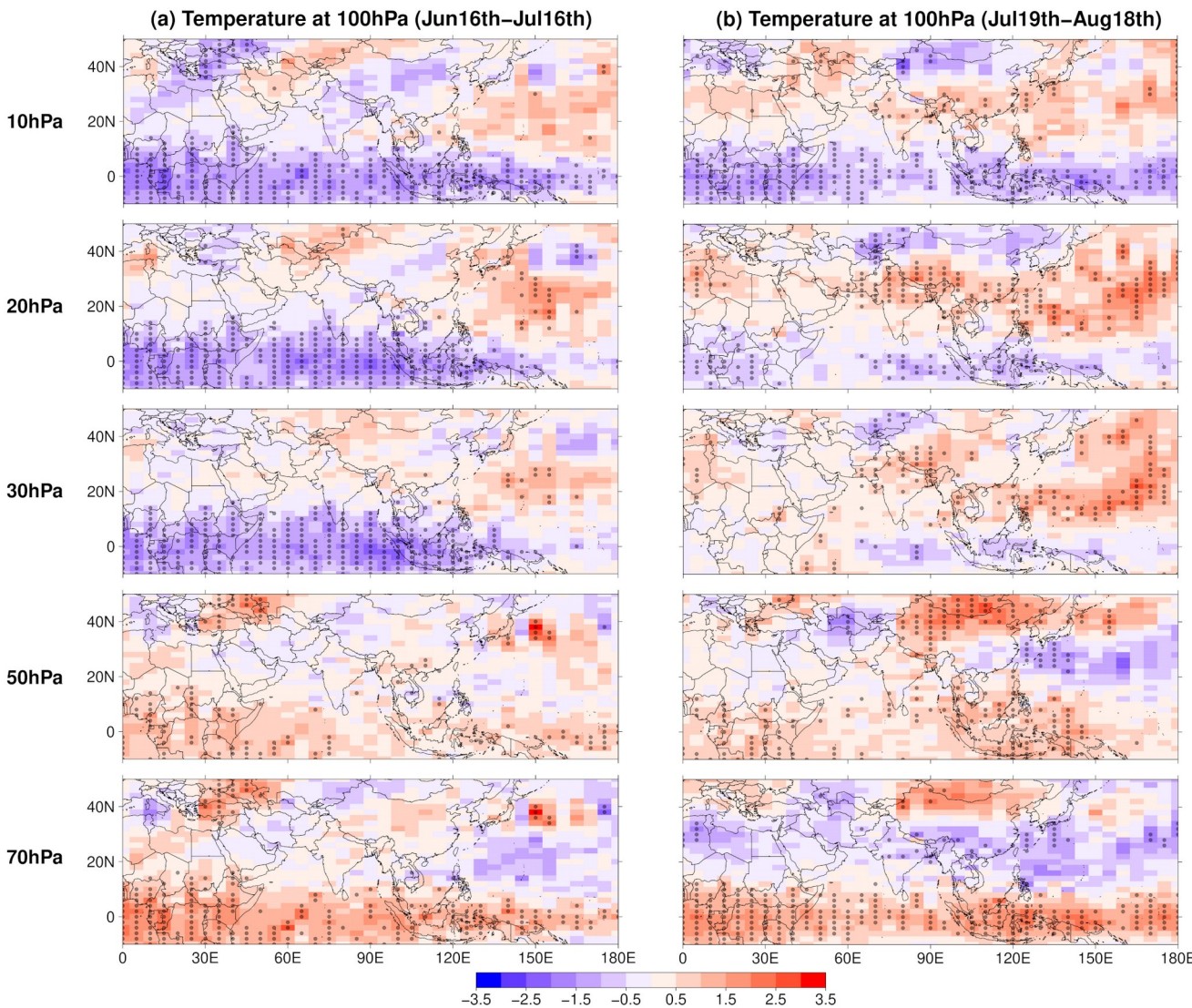

**Figure A3: QBO-W minus QBO-E differences for MLS temperature at 100hPa averaged over the 31-day period between 16 June and 16 July (a) and between 19 July and 18 August (b), over 2005-2020. Each row corresponds to a different level at which the QBO phases were defined from 10hPa (top row) to 70hPa (bottom row). Dots indicate significance at the 95% confidence level.**

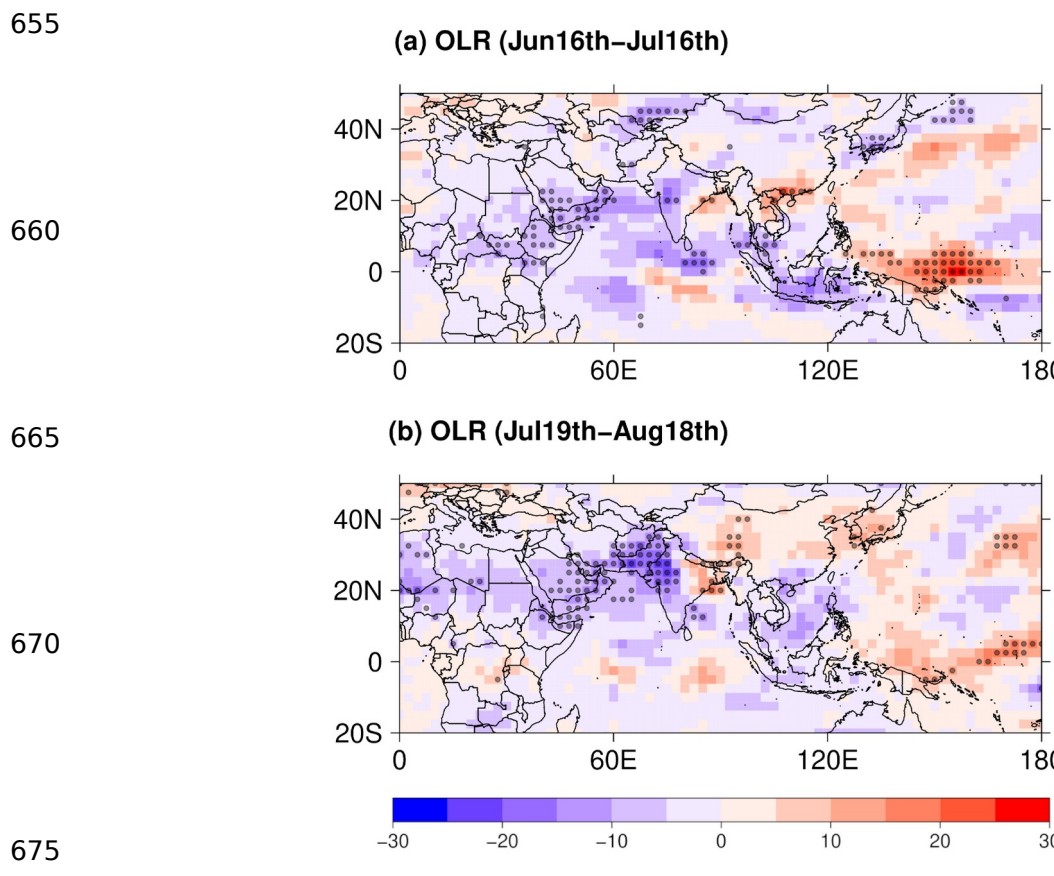

**Figure A4: QBO-W minus QBO-E differences for OLR averaged between June 16th and July 16th (a) and between July 19th and August 19th (b). In (a) differences correspond to the QBO index for July defined at 10hPa while for (b) we chose the QBO index defined at 20hPa for August. Dots indicate significance at the 95% confidence level.**

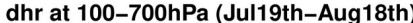

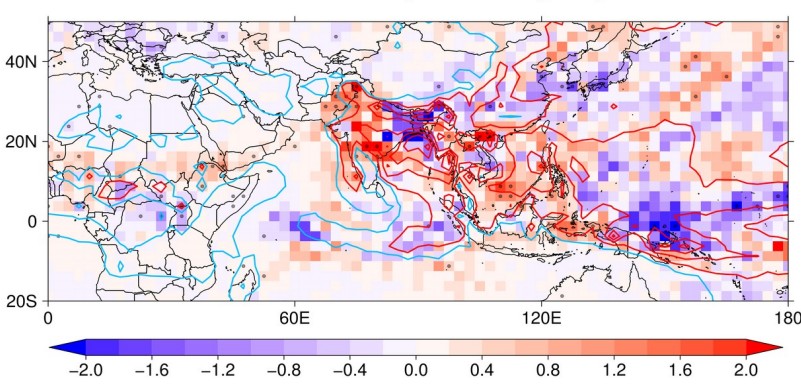


**Figure A5: QBO-W minus QBO-E differences for diabatic heating rate (color shades) averaged between 100hPa and 700hPa over the period between 19th July and 18th August. Blue/red contour lines show negative/positive anomalies of the temperature eddy field with contour intervals at every 1K/day starting at 1K/day and -1K/day for positive and negative anomalies respectively.Dots indicate significance at the 95% confidence level.**



*Data availability.* MLS H$_2$O and Temperature version 4.2 data can be obtained from the MLS website https://mls.jpl.nasa.gov. The ERA5 data can be accessed through the Copernicus Climate Data Store website 695 https://cds.climate.copernicus.eu/cdsapp#!/home.

*Author contributions.* CP and NP performed the data analysis. CP, NP, FP and DG contributed to the discussion of results. CP wrote the text. NP, FP and DG made the final review.

*Competing interests.* The authors declare that they have no conflict of interest.

*Acknowledgements.* This research was funded by the Spanish Ministerio de Economía y Competitividad through the project 700 Variabilidad del Vapor de Agua en la Baja Estratosfera (CGL2016-78562-P).

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
