# Peer review of "QBO modulation of stratospheric water vapour in the Asian monsoon"

_EGUsphere, 2023_

## Author Comment (AC1)

Response to reviewer 1:

(1) The use of ERA5 "fraction of cloud cover" data

 The authors use vertically resolved "fraction of cloud cover" data from ERA5 as a proxy of, in the end, tropical deep convection that provides source of large-scale diabatic heating to produce equatorial Kelvin and Rossby wave response (or the Matsuno-Gill response). First, reanalysis cloud data are product of forecast model, without observations assimilated, and thus in general less reliable compared to e.g. NOAA OLR data. Second, the authors focus on the region 150-100 hPa for these cloud data and discuss deep convection, but the clouds in this altitude region are primarily cirrus and anvil clouds, and may not be directly related to the core of deep convection that results in large-scale diabatic heating to produce the Matsuno-Gill type response. Because authors' discussion very heavily relies on detailed structure/distribution in ERA5 fraction of cloud cover data at 150-100 hPa, and because (as discussed below) the features that the authors point out and emphasize are not very clear to me, I started to wonder whether the chosen cloud data set is appropriate or not.

We do indeed share the view that ERA5 fraction of cloud cover data may be less reliable than NOAA OLR data, so in the manuscript we add figure A2 in the appendix showing QBO-W minus QBO-E differences for NOAA OLR averaged between June 16th and July 16th (a) and between July 19th and August 19th (b). This figure serves to verify the signal observed in the "fraction of cloud cover" that we show in figures 6 and 7, from an independent observational data set. Also, in the new version of the manuscript we incorporate the analysis of the diabatic heating rate of ERA5. These new analyses and data corroborate the patterns and changes found in the ERA5 fraction of cloud cover.

It is true that our analysis does not allow us to determine which cloud types are affected by the QBO. Cirrus are the most common clouds at levels between 200 and 100hPa and, therefore, it is very likely that the QBO signal is mainly modulating the presence of this type of clouds. This result is consistent with Sweeney et al. (2023), who show that the QBO primarily affects cloudiness between 100hPa and 200hPa, mainly impacting cirrus clouds. However, also this study shows evidence of the QBO signal on cloudiness between 100hPa and 200hPa associated with opaque clouds, which are clouds often associated with deep convection and thick anvil. In fact, Giorgetta et al. (1999) argue that QBO acts primarily by raising the height of

cloud tops, which, when the QBO causes a cooling of the tropopause, can more frequently reach levels between 100hPa and 150hPa rather than between 150hPa and 200hPa.

The possibility that changes in cloudiness in the upper troposphere may cause changes in circulation and excitation of wave trains was addressed in Slingo and Slingo (1990), Giorgetta et al. (1999) and Peña-Ortiz et al. (2019). These studies show that diabatic heating caused by cloudiness at the upper troposphere, including that associated with tropical cirrus clouds, can excite wave trains that can propagate to higher latitudes. Slingo and Slingo (1990) found that the major dynamical response to upper tropospheric diabatic heating is restricted to the tropics and the subtropical jets and that this response is characterized by an anticyclonic dipole to the north and south of the diabatic heating, with the possibility to excite wave trains propagating to higher latitudes.

We consider that this discussion is relevant and therefore, in the new version of the manuscript we show how the QBO signal on the diabatic heating rate in the upper troposphere is consistent with the signal in the cloudiness and, in turn, consistent with the observed response in the circulation which is characterized by the generation of divergence in the zonal wind with winds from the east to the west of the diabatic heat release zone and winds from the west to the east of it. At the same time, as described by Slingo and Slingo (1990), anticyclonic gyres can also be distinguished to the north and south of this region, being more intense to the north as the heat release is also displaced to the north.

However, we agree with the referee that the complete structure, typical of Matsuno Gill response, is not so clearly distinguishable and therefore in the revised manuscript  we will not refer to it. It is also true that to demonstrate a cause-effect relationship between the diabatic heating and the subsequent generation of the wave train, a dynamic model would be necessary and, in the new version, we make it clear that this is an hypothesis suggested by the results but that its demonstration would require simulations that are beyond the scope of this article.

(2) The Matsuno-Gill response

 In Section 4, in Figures 6-8, the authors show the ERA5 cloud data at 125-150 hPa and ERA5 temperature and wind anomalies, and mention that these are the Matsuno-Gill response. While it is well known that if we give diabatic heating at the equator or slightly off-equatorial region, we see the so-called Matsuno-Gill response in the wider regions of the tropical-to-subtropical

atmosphere including the tropopause region, it is not clear to me in the current specific cases which group of deep convection, shown on the figures, is actually responsible for the specific 100 hPa temperature and wind anomalies over the Asian monsoon region. The authors need to clearly show the heating-response pair for each set of figures, and to show somehow (using e.g. a very simple model) the justification that they are actually the pairs.

In the new version of the manuscript, the use of diabatic heating has allowed us to more clearly delineate the regions where this heat is released and the circulation anomalies that respond to this heating. However, it is true that the Matsuno-Gill response is not so clearly distinguishable, primarily because it is difficult to distinguish the response in terms of a Kelvin wave to the east of the heating region. It is possible that either the heat released is not sufficient to generate this structure, or the area where positive latent heat release anomalies are observed sometimes appears to form a dipole next to a region where anomalies of the opposite sign are observed, preventing the Rossby-Kelvin pair from developing.

Studies by Slingo and Slingo (1991) and Giorgetta et al. (1999) that address the dynamical response to diabatic heating of the upper troposphere describe this response in terms of the formation of anticyclonic gyres to the north and south of the heating region and the excitation of wave trains. In the new version of the manuscript we emphasise these structures, which are clearly observable, and are more careful with references to the Matsuno-Gill response. In particular, we discuss that our analysis allows only the identification of consistency in the patterns with the Matsuno-Gill response but that for proving a causal relation a mechanistic model would be needed (see our answer to the previous question).

(3) The choice of QBO indices

 The authors used monthly mean Singapore zonal wind data at 10 hPa, 20 hPa, 30 hPa, 50 hPa, and 70 hPa for (potential) QBO indices. While the authors' approach is understandable as first trials, I think that the final choice should be made in terms of direct relevance to the current problem. What we need here is e.g. vertical displacement, or temperature anomaly, or temperature gradient (static stability) anomaly around e.g. the tropopause over the tropics and over the Asian monsoon region due to the QBO and/or its secondary circulation. In other words, please explain the relevance (or the phase relationship) to the tropopause-level variables of the 10 hPa Singapore winds for July and the 20 hPa Singapore winds for August.

We fully agree that the aspects explaining the QBO signal on water vapour in the Asian Monsoon UTLS are related to the QBO signal on temperature and circulation at the tropopause level over the tropics and over the Asian monsoon region. However, Figures 1 and 2 show that the signal over the Monsoon water vapour is stronger when we define the QBO phases as a function of the zonal wind at 10hPa or 20hPa for July and August respectively. This signal is, of course, not due to what happens at these levels of the stratosphere but due to the impact of the QBO at the tropopause and its impact on temperature and circulation at this same level.

In fact, in the case of July, the water vapour signal for the QBO defined at 10hPa is practically the same but with the opposite sign to the one observed in the last row of figure 1, for the QBO defined at 70hPa. However, because the QBO signal over the zonal wind weakens in the lower stratosphere, the use of levels between 70hPa and 100hPa to define the QBO phases can be problematic and significantly reduce the number of cases.

We chose to show the signal over water vapour and temperature at 100hPa for the QBO defined at different levels to show the relationship between the signals over these two variables. However, in the new version we have simplified figures 1, 2 and 4 to show only the panels corresponding to the QBO phases defined at 10hPa and 20hPa for July and August respectively. In the revised manuscript, we will give a more detailed description of the characteristics of the QBO signal on temperature and circulation in the UTLS during these phases.

(4) The QBO secondary circulation, and then the potential Matsuno-Gill response

(This may be more like a comment, not a strong suggestion.) To me, it is more logical that the (zonal mean) QBO secondary circulation is first explained and analyzed, and then the anomalous Asian monsoon region is pointed out. Then, the tropical convection anomalies are analyzed in the Indian-Ocean and Indonesian sector. Then, the potential link of those convection anomalies to the Asian monsoon region through the Matsuno-Gill response is proposed.

The exact latitude where the subtropical part of secondary circulation maximizes might not be very clear in the past works, but the following paper may be a good starting point:

Hitchman et al., 2021, https://doi.org/10.2151/jmsj.2021-012

The one for specific months needs to be analyzed by the authors (and actually shown in the manuscript).

We share with the reviewer the need to illustrate more clearly the behaviour of the secondary meridional circulation of the QBO (zonal average) and then point out the anomalies over the Asian monsoon region. For this we will unify Figures 5 and A1 and expand the description of the secondary meridional circulation. We believe that this will allow a better understanding of the significance of the anomalies found over the monsoon.

Furthermore, it is not very clear to me what is the final process that mainly controls the water vapor in the Asian monsoon anticyclone. Is it local dehydration in association with the temperature anomalies or the wet/dry air transport changes in association with the wind anomalies, when the authors discuss the Rossby-wave part of the Matsuno-Gill response?

Our results, beyond showing the QBO signal in water vapor over the Asian Monsoon, attempt to explain this signal from the temperature anomalies associated with the QBO in the UTLS. In the case of water vapor in August, our results show that the QBO has a significant impact on temperature on the southern flank of the monsoon from mid-July to mid-August that modulates, through local dehydration, the water vapor content of the monsoon. Results suggest that in this region the temperature anomaly associated with the secondary circulation intensifies due to the wave train generated by the impact of the QBO on tropical convection in the previous weeks. Thus, this wave train is characterized, during QBO-W and with respect to QBO-E, by a cyclonic anomaly on the southeast flank of the AM that inhibits convection and favors subsidence in this region, generating a warm anomaly in July/August that intensifies the warming associated with the secondary circulation of the QBO and causes a moistening of the AM.

However, with respect to water vapor in July, our results show that it is the temperature over the tropical Indian Ocean that controls the interannual variability of water vapor in the monsoon. In fact, the QBO signal on monsoon water vapor in July is in phase with the signal in the equatorial UTLS. Thus a cooler and drier tropical UTLS coincides with a drier monsoon. Moreover, in this case, the QBO signal on temperature on the southern flank of the monsoon is very weak. It is therefore possible, as suggested in the manuscript, that the transport to the Asian monsoon of anomalously dry or moist air from the tropical Indian, is contributing to the signal that we find over the monsoon. In any case, in order to determine the separate effects of transport and

temperature anomalies, we would need to analyze the trajectories of the air masses using a Lagrangian model, which is out of the scope of this manuscript. In the new version of the manuscript we will try to clarify this issue and also to explicitly discuss the limitations of our analysis.

(5) Different processes are proposed for July and August

 Based on the analysis results, the authors suggest that different processes are operating in the month of July and August at the QBO time scales. I am not fully convinced whether this is possible/reasonable. If this were true, the seasonal progress of the Asian summer monsoon should be quite robust in each year, and the seasonal features are clearly different between July and August. Or, do I misunderstand something?

Our results reveal that indeed the QBO signal over water vapor has different characteristics in intensity and sign between July and August. While in July it is weaker and in phase with  water vapor anomalies over the equator, in August it is stronger and shows a lag in relation to the signal over the equatorial moisture. One of the keys to this difference lies in the different impact of the QBO on the temperature of the southern flank of the monsoon (a region whose temperature is considered key to the modulation of monsoon vapor content) during Jun/Jul (mid-June to mid-July) and Jul/Aug (mid-July to mid-August). As explained in the answer to the previous question, in Jul/Aug significant temperature anomalies are observed over this region, which are consistent with changes in monsoon water vapor content. However, in Jun/Jul, temperature anomalies over this area are very weak, and therefore the monsoon vapor content seems to depend more on the temperature and moisture of the air over the equatorial region (figure 3a of the manuscript).

Results suggest that the seasonal evolution of the impact of the QBO on cloudiness in the tropical upper troposphere may explain these differences. Thus we see that during Jun/Jul there is diabatic warming, associated with QBO-W and with respect to QBO-E, over the tropical Indian Ocean which generates an anticyclonic gyre over India and a cooling observed in the temperature eddy field. This cooling is partially compensated by the warm anomalies of the zonal average (associated with the secondary meridional circulation of the QBO shown in figure A1), so that the actual temperature field shows very weak cold anomalies that have a limited impact on the water vapor content of the monsoon. However, in July/August, the QBO does not significantly modulate the cloudiness over the tropical Indian Ocean and thus there is not an

intensification of the tropical easterly wind over this region and, as a consequence, the anticyclonic anomaly over India that appeared in June/July does not occur. Instead, we observe a wave train that is characterized by a cyclonic gyre over India, which inhibits cloudiness and generates a warm anomaly in this region, which, coupled with the warming of the secondary circulation of the QBO, causes a moistening of the AM during QBO-W relative to QBO-E. This is consistent with results of Giorgetta et al. (1999), which show that the latent heat released in June gives rise to a wave train observed in Jul, this wave train could be the result of the modulation of the upper tropospheric cloudiness during Jun/Jul, which in Jul/Aug gives rise to a wave train with the characteristics previously described. Furthermore, the evolution over the summer of the impact of the QBO on cloudiness in the tropics described in our study is also in agreement with Sweeney et al. (2023), which found that the strongest cloud fraction response to the QBO occurs in boreal spring and early summer, peaking during (May, June and July).

The  evolution over the summer of the response to QBO of diabatic heating rate and circulation can be seen in figure 1 of this document. In this figure the differences between QBO-W and QBO-E have been calculated in all cases by defining the phases from the Singapore index for July at 10hPa. In this way we can follow the evolution of the QBO signal from mid-June (June 10th- July 10th, top row) to mid-August (July 19th- August 18th).  This figure shows how, since the beginning of the summer, a diabatic warming has been observed in the equatorial region, reaching its maximum over the Indian Ocean, south of the Bay of Bengal. In fact, this warming is already observed throughout the month of June (not shown). This latent heat release increases in intensity until it reaches its maximum during the month of July (July 1st-July 31st) and then weakens rapidly. Associated with this heat release and only while it is occurring (until the period July 1st- July 31st), a circulation response is found, which is characterized by an acceleration of easterly winds to the west of the heat release region and the formation of an anticyclonic gyre over India, centered at 20N. This anticyclonic gyre intensifies the climatological anticyclone that dominates this region and, therefore, air rising motions, generating a cold anomaly associated with adiabatic cooling. It can also be observed how this anticyclone is part of a wave train that propagates northwards and southwards in the northern and southern hemispheres. As the diabatic warming weakens (last two rows in figure 1), the easterly wind anomaly over the equator also weakens and the wave train evolves so that the anticyclonic gyre over India gives way to a cyclonic gyre and a warm anomaly over the southern slack of the monsoon for the period Jul 19-Aug 18 (bottom row).

In the manuscript, and in order to optimize the QBO signal over the monsoon water vapor, we have made the analysis of the signal in July and August using a QBO index different from each month since in the case of July (August) we used July (August) winds. This has the counterpart that, in our case, the set of July and August months defined as QBO-E or QBO-W are not consecutive, they do not belong to the same years, and therefore the evolution of the signature over the summer is not straightforward. In any case, in the revised version of the manuscript we will include a figure (an improved version of figure 1 of this document) to show the evolution of the QBO signal on diabatic heating, circulation and temperature over the summer. We will also add an explicit analysis of the diabatic heating rates and will extend the latitudinal range in the figures showing the circulation response so that the generated wave trains can be better appreciated.

**Wind & T eddy fields at 100hPa**

**Diabatic heating rate at 100hPa**

[Figure]

June 10th-July10th

June 16th-July16th

July 1st-July31st

July 10th-Aug9th

July 19th-Aug18th

Figure 1: QBO-W minus QBO-E differences (phases defined from the July Singapore wind at 10hPa) for the eddy fields of zonal wind (color shades), horizontal wind (arrows) (b) and temperature (contours) (left column) and for diabatic heating rate (right column) averaged over the period between 10 June and 10 July, 16 June and 16 July, 1 July and 31 July,  10 July and 9 August and 19 July and 18 August. Blue/red contour lines show negative/positive anomalies of the temperature eddy field with contour intervals at every 0.5K starting at 0.25K and -0.25K for positive and negative anomalies respectively. Dots indicate significance at the 95% confidence level for zonal wind.

This leads me to the question (1). The features shown in the manuscript might be heavily dependent on the data set used, and ERA5 100-150 hPa fraction of cloud cover data might not be appropriate to be used as the observation of convective heating.

We agree that ERA5 fraction of cloud cover data is less reliable than NOAA OLR data (see the response to the first question). Thus, in the first version of the manuscript we already included the OLR analysis to verify the observed anomalies in the fraction of cloud cover. Additionally,  in the revised version we have also added the analysis of the diabatic heating rate of ERA5, showing how the QBO signal on the diabatic heating rate in the upper troposphere is consistent with the signal in the cloudiness and, in turn, consistent with the observed response in the circulation.

References

Giorgetta, M. A. and Bengtsson, L.: Potential role of the quasi-biennial oscillation in the stratosphere-troposphere exchange as found in water vapor in general circulation model experiments, J. Geophys. Res., 104, 6003–6019, https://doi.org/10.1029/1998jd200112, 1999.

Randel, W. J., Moyer, E., Park, M., Jensen, E., Bernath, P., Walker, K., and Boone, C. (2012), Global variations of HDO and HDO/H2O ratios in the upper troposphere and lower stratosphere derived from ACE-FTS satellite measurements, *J. Geophys. Res.*, 117, D06303, doi:10.1029/2011JD016632.

Slingo, J.M. and Slingo, A. (1991), The response of a general circulation model to cloud longwave radiative forcing. II: Further studies. Q.J.R. Meteorol. Soc., 117: 333-364. https://doi.org/10.1002/qj.49711749805

Sweeney, A., Fu, Q., Pahlavan, H. A., & Haynes, P. (2023). Seasonality of the QBO impact on equatorial clouds. Journal of Geophysical Research: Atmospheres, 128, e2022JD037737. https://doi.org/10.1029/2022JD037737

Ueyama, R., Schoeberl, M., Jensen, E., Pfister, L., Park, M., & Ryoo, J.-M. (2023). Convective impact on the global lower stratospheric water vapor budget. *Journal of Geophysical Research: Atmospheres*, 128, e2022JD037135. https://doi.org/10.1029/2022JD037135

Ueyama, R., Jensen, E. J., & Pfister, L. (2018). Convective influence on the humidity and clouds in the tropical tropopause layer during boreal summer. *Journal of Geophysical Research: Atmospheres*, 123, 7576–7593. https://doi.org/10.1029/2018JD028674

---

## Author Comment (AC2)

Response to reviewer 2:

Review for 'QBO modulation of the Asian Monsoon water vapour' by Peña-Ortiz et al.

This work studies the mechanisms of how Quasi-Biennial Oscillation (QBO) will influence the tropical tropopause layer water vapor by regulating temperature and convection. This work focuses on the Asian Monsoon (AM) region, a region where it is important while there remains a lot of uncertainty in the water vapor budget. By analyzing observational data, the authors explain how QBO will influence the water vapor from a dynamics perspective, filling current knowledge gaps. Most of the analytical work in this paper is clear and logical, and the paper is well-written. I recommend the paper be accepted after considering the following suggestions.

Major comments:

1.   It would be helpful for readers to better understand the processes involved if the authors include figures showing the climatology, QBO-w mean, and QBO-e means of water vapor, temperature, outgoing longwave radiation (OLR), wind fields. This additional visual representation could enhance comprehension.

Certainly, we agree on the need to show the climatological values for vapor, temperature, wind and OLR for a better understanding of the results and we are going to do so for the revised version of the manuscript in which these values will be added to the anomaly figures or, in some cases, figures will be added to show them. In the case of the mean values for QBO-E and QBO-W, because the anomalies associated with each of the QBO phases are small compared to the climatology, we think that they do not add additional information with respect to the climatology and so we have not added them.

2.   Table 1 indicates the number of QBOe and QBOw cases but doesn't provide information on their strength. With a sample size of 5-9 cases, there remains substantial uncertainty in the results. For instance, the differences observed between July and August may be influenced by variations in QBO strength due to the limited record. To address this, consider:
a.   When discussing the relationship between water vapor and temperature, use MLS water vapor data, and temperature over 2005-2020.

b.    When discussing how QBO modulates the deep convection and thus temperature, use a longer record (ERAi, NOAA OLR, and Singapore QBO data all have a longer record).

c.    Instead of counting the number of QBO events, consider presenting variables like wind (U) or the correlation between U and H2O to strengthen your analysis.

Although it is true that the availability of high-quality data for water vapor significantly limits the period of study (2005-2020), we believe that it contains a sufficient number of QBO-W and QBO-E cases (between 9 and 6) to perform a meaningful analysis. In our opinion,extending the study period including less reliable water vapor data from the 80s and 90s would not improve the significance of the results. On the other hand, the quality of the temperature, wind, OLR or diabatic heating rate data at UTLS levels can also be compromised as we go back in time.

In order to assess possible variations in the QBO strength for the QBO-W and QBO-E cases used in our study, figure 1 compares the QBO amplitude at different pressure levels according to the values of the Singapore sonde monthly zonal wind for QBO-W and QBO-E defined at 10hPa and 20hPa for July and August for the same time period used in the paper, 2005-2020, and following the same procedure for the phases definitions. The figure shows a slightly higher amplitude of the QBO at UTLS levels during July compared to August, and this may indeed contribute to the stronger QBO signal on temperature and cloudiness during that month. In fact, this is consistent with Sweeney et al. (2023), who show that deep intrusions of zonal wind anomalies into the upper troposphere are only visible during boreal spring and early summer, peaking during MJJ. They found that, in a synchronized way, QBO temperature anomalies penetrate to the lowest altitudes and the response of the cloud fraction reaches its maximum and extends  from above tropopause to ꟷ12.5 km. Although this study used observational data for the period 2006-2020, they found, using ERA5 reanalysis, a similar seasonality of the zonal wind response to the QBO over the period 1979-2020. In the same way, figure 2 of the present document, which shows latitude-height cross sections of QBO-W minus QBO-E differences for ERA5 zonal mean temperature and zonal wind averaged for June 16th - July 16th and July 19th - August 18th for the period 1980-2020, also evidences slightly stronger QBO anomalies in the UTLS during Jun-Jul than during Jul-Aug. This would support the idea that the difference in the amplitude of the QBO signal in the zonal wind in the UTLS (and synchronously in all other variables) between July and August found for the period 2005- 2020 is not an artifact associated with the limited number of cases, but something related to the seasonality of the QBO strength

in the UTLS. In any case, we consider this to be an obvious limitation of our work and will explain it in the new version of the manuscript.

[Figure]

Figure 1: QBO-W minus QBO-E differences for the Singapore sonde monthly zonal wind at different pressure levels for QBO-W and QBO-E defined at 10hPa and 20hPa for July and August for 2005-2020. QBO phases defined at 20hPa/10hPa are shown as dashed lines/solid lines while values for August/July are shown in red/blue colors.

[Figure]

Figure 2: Latitude-height cross sections of QBO-W minus QBO-E differences for ERA5 zonal mean temperature (colors) and zonal wind (black contours) for June 16th - July 16th (top row) and July 19th - August 18th (bottom row) and for the period 1980-2020. QBO phases were defined at 10 hPa using the Singapore index for July (top left) and August (bottom left) and at 20hPa using the Singapore index for July (top right) and August (bottom right). Solid/dashed contour lines show positive/negative anomalies with contour intervals at every 2m/s from 1m/s/-1m/s for positive/negative anomalies. Dots indicate significance at the 95% confidence level.

Finally, we have performed some checks in order to test the sensitivity of our results to the period of study. Thus, to test whether the observed differences in temperature anomalies on the southern flank of the monsoon between Jun-Jul and Jul-Aug for the period 2005-2020 are also observed during the extended period 1980-2020, figure 3 depicts latitude-height cross sections of QBO-W minus QBO-E differences for ERA5 temperature and zonal wind averaged over 60E-120E and between June 16th and July 16th (a) and between July 19th and August 18th (b) for the period 1980-2020. Thus, it is equivalent to figure 4 in the manuscript but for a 41-year period. Consistent with results in the manuscript for 2005-2020, this figure shows no significant temperature anomalies in the UTLS (around 100hPa) over latitudes corresponding to the southern flank of the monsoon (20N-30N) during Jun/July while a significant warming, although

weaker, of this region is observed during Jul/Aug. We mention the robustness of this result in the revised manuscript.

In any case, as we have said, we understand that the limited extension of the study period is an obvious shortcoming of this work and we will discuss this in the revised version of the manuscript. We will also add the information about the amplitude of the QBO at each level in table 1.

[Figure]

Figure 3: Latitude-height cross sections of QBO-W minus QBO-E differences for ERA5 temperature (colors) and zonal wind (black contours) averaged over 60E-120E and between June 16th and July 16th (a) and between July 19th and August 18th (b) for the period 1980-2020. In (a) differences correspond to the QBO index for July defined at 10hPa while for (b) we chose the QBO index defined at 20hPa for August. Solid/dashed contour lines show positive/negative anomalies with contour intervals at every 2m/s from 1m/s/-1m/s for positive/negative anomalies. Dots indicate significance at the 95% confidence level.

3.    The paper currently discusses the choice of different QBO levels extensively, which may distract from the main topic. The conclusion that 10 hPa/20 hPa is the best proxy may be sample-size dependent and not universally applicable so can be removed from the result section. Consider consolidating this discussion with Table 1 and relocating it to Section 2, allowing Sections 3 and 4 to focus on scientific results.

In line with this comment we have simplified figures 1, 2 and 4 to show only those panels corresponding to the QBO phases defined at 10hPa for July and 20hPa for August and have moved the previous figures with all panels to the appendix.  In this way we have also been able to simplify the text and reduce the discussion regarding the signal in the vapour and the

temperature as a function of level for the definition of the QBO phases. Following the reviewer's recommendation and, as far as possible, we have moved the discussion on the dependence of the QBO signal in tropopause on the level at which the phases are defined to section 2, "Data and Methodology".

4.  Need more discussion on using OLR as a proxy of the deep convection. Although Randel et al. (2015) conclude that deep convective cooling plays a main role in the water vapor budget over AM and North American regions, there are also many recent studies using radar with a much finer resolution to observe deep convection and overshooting over the North American regions, and conclude that the deep convection impact is moistening (Chang et al., 2023; Smith et al., 2017; Tinney & Homeyer, 2021; Yu et al., 2020). Asian monsoon region is different from the NA region and there is no very good coverage radar product over the AM region to test convective moistening/drying, so convective impact still could be cooling over this region. However, a discussion on the shortcoming of using OLR as a proxy is necessary.

In our study we observed that the modulation of cloudiness by the QBO plays a dual role. On the one hand, cloud modulation over the equatorial region contributes to a response in circulation and temperature over the southern flank of the monsoon that modulates the water vapor content of the monsoon. In this case, changes in equatorial cloudiness cause changes in diabatic heating and thus generate the dynamic response. To add further evidence to this result, in the revised version we have added the analysis of the ERA5 diabatic heating rate, which allows us to verify that changes in tropical cloudiness, even when occurring in the UTLS, are likely to cause the release of latent heat, which is essential to generate a response in the circulation.

On the other hand, changes in the circulation over the southern flank of the monsoon cause changes in convection and temperature over this region. In this case, we have only considered the impact of convection on temperature but have not analyzed the possible impact of direct water vapour injection over the monsoon. While it is true, as suggested by the reviewer, that several studies have pointed to the moistening effect of deep convection in monsoon regions, particularly over the North American monsoon, the contribution of these convective systems to the overall budget of water vapour transport over monsoon regions is still questionable (Ueyama et al. 2023). For example, while Ueyama et al. (2018) showed that deep convection within the Asian monsoon anticyclone can enhance water vapour in the lower stratosphere by 30%, Plaza et al. (2021) found no contribution from this mechanism despite using a similar approach.

Furthermore, the hierarchy of processes controlling the water vapour signal in monsoon regions may be monsoon dependent due to differences in thermodynamic structure between the Asian and North American monsoons. For example, Randel et al. (2012) found a maximum of δD only for the North American monsoon as evidence for different magnitudes of transport processes playing a role in monsoon water vapour signals. Therefore, it is not possible to directly extrapolate to the case of the Asian monsoon the conclusions drawn in the scientific literature for the observed wetting of the North American monsoon in relation to convection. We agree with the reviewer that to attend the full impact of convection within the Asian Monsoon region, it would be necessary to obtain new and more accurate observations of this region in addition to OLR. In this study we do not aim to quantify this impact and the causal links between the changes in temperature and cloud fraction in the Asian Monsoon region related to QBO phases should be further studied using a dynamical model for this purpose.

In the revised version, we will expand the discussion on this issue and explicitly clarify the limitation in determining the role of convection, not having considered its possible impact through direct water vapor injection.

Specific comment:

1.    After first introducing the region, consider adding latitude information (e.g., Indian, tropical Indian Ocean).

Latitude information will be added in the revised version.

2.    Line 16-18: The sentence is long and slightly challenging to comprehend. Consider breaking it into two or more sentences for clarity.

Corrected in the revised version.

3.    Figure 3: This figure could benefit from subtitles and adding a significant level to figures b and d. Additionally, consider adding a second x-axis with lag dates to improve readability.

Corrected in the revised version.

4.    Figure 5: The dots in this figure are unclear when overlapping with the wind field.

Corrected in the revised version.

5.     Line 205: It's unclear how this paragraph differs from the one starting at line 140, as they both discuss Figure 3. Please clarify the distinction.

 The dots have been enlarged and are more clearly distinguishable in the revised version of the figure.

6.     Line 289: The sentence is unclear. Isn't the secondary circulation over the mid-latitudes, and this paragraph is discussing the equatorial region? Could you please explain more?

The secondary meridional circulation (SMC) is characterized by a sinking motion at the equator in westerly shear zones and rising in easterly shear areas. A maximum (minimum) in temperature at the equator in westerly (easterly) shear zones is necessary to maintain the thermal wind balance. The sinking (rising) motions produce adiabatic heating (cooling) to preserve positive and negative temperature anomalies against thermal damping (Baldwin et al., 2001). As Plumb and Bell (1982) illustrated in their 2-D model, the SMC is also distinguished by meridional convergence (divergence) zones over the equator coinciding with the location of maximum westerly (easterly) wind. The QBO temperature anomaly changes sign at approximately ±15° owing to rising (sinking) motions that compensate the sinking (rising) motions at the equator (Baldwin et al., 2001; Choi et al., 2002).

In the revised version, we will clarify that in line 289 we are referring to the equatorial branch of the SMC.

7.     Line 359: In the sentence, "and a secondary meridional circulation of the QBO...," Is the secondary circulation transport the equatorial temperature anomaly?

What we intend to say in that sentence is that the warm temperature anomaly appears as a combination of two effects. The first one is the effect of the wave train observed between mid-July and mid August, characterized by a cyclonic gyre over the southeast flank of the Monsoon. This cyclonic gyre involves a weakening of the climatological rising motions (a downward anomaly) and also a weakening of the cloudiness northeast of India, which causes a warming over this region. This warming is observed in the eddy temperature field (corresponding to the total field minus the zonal mean) as well as a warming that is enhanced by the warm

temperature anomaly associated with the secondary meridional circulation. This anomaly contributes to the Monsoon moistening.

We will clarify this sentence in the revised version.

Reference

Chang, K.-W., Bowman, K. P., & Rapp, A. D. (2023). Transport and Confinement of Plumes From Tropopause-Overshooting Convection Over the Contiguous United States During the Warm Season. Journal of Geophysical Research: Atmospheres, 128(2), e2022JD037020. https://doi.org/10.1029/2022JD037020

Smith, J. B., Wilmouth, D. M., Bedka, K. M., Bowman, K. P., Homeyer, C. R., Dykema, J. A., Sargent, M. R., Clapp, C. E., Leroy, S. S., Sayres, D. S., Dean-Day, J. M., Paul Bui, T., & Anderson, J. G. (2017). A case study of convectively sourced water vapor observed in the overworld stratosphere over the United States. Journal of Geophysical Research: Atmospheres, 122(17), 9529–9554. https://doi.org/10.1002/2017JD026831

Tinney, E. N., & Homeyer, C. R. (2021). A 13-year Trajectory-Based Analysis of Convection-Driven Changes in Upper Troposphere Lower Stratosphere Composition Over the United States. Journal of Geophysical Research: Atmospheres, 126(3), e2020JD033657. https://doi.org/10.1029/2020jd033657

Yu, W., Dessler, A. E., Park, M., & Jensen, E. J. (2020). Influence of convection on stratospheric water vapor in the North American monsoon region. Atmospheric Chemistry and Physics, 20(20), 12153–12161. https://doi.org/10.5194/acp-20-12153-2020

References

Giorgetta, M. A. and Bengtsson, L.: Potential role of the quasi-biennial oscillation in the stratosphere-troposphere exchange as found in water vapor in general circulation model experiments, J. Geophys. Res., 104, 6003–6019, https://doi.org/10.1029/1998jd200112, 1999.

Randel, W. J., Moyer, E., Park, M., Jensen, E., Bernath, P., Walker, K., and Boone, C. (2012), Global variations of HDO and HDO/H2O ratios in the upper troposphere and lower stratosphere derived from ACE-FTS satellite measurements, *J. Geophys. Res.*, 117, D06303, doi:10.1029/2011JD016632.

Slingo, J.M. and Slingo, A. (1991), The response of a general circulation model to cloud longwave radiative forcing. II: Further studies. Q.J.R. Meteorol. Soc., 117: 333-364. https://doi.org/10.1002/qj.49711749805

Sweeney, A., Fu, Q., Pahlavan, H. A., & Haynes, P. (2023). Seasonality of the QBO impact on equatorial clouds. Journal of Geophysical Research: Atmospheres, 128, e2022JD037737. https://doi.org/10.1029/2022JD037737

Ueyama, R., Schoeberl, M., Jensen, E., Pfister, L., Park, M., & Ryoo, J.-M. (2023). Convective impact on the global lower stratospheric water vapor budget. *Journal of Geophysical Research: Atmospheres*, 128, e2022JD037135. https://doi.org/10.1029/2022JD037135

Ueyama, R., Jensen, E. J., & Pfister, L. (2018). Convective influence on the humidity and clouds in the tropical tropopause layer during boreal summer. *Journal of Geophysical Research: Atmospheres*, 123, 7576–7593. https://doi.org/10.1029/2018JD028674

---

## Author Response (AR2)

Second Review report on "QBO modulation of stratospheric water vapour in the Asian monsoon" by Cristina Peña-Ortiz et al.

The authors made substantial revisions in response to my comments. The authors now do not explicitly refer to "the Matsuno-Gill pattern". This is probably a reasonable approach, avoiding unnecessary confusion by the readers (i.e. which "pattern" the authors are referring to?). But, a drawback may be that the authors' idea of response to tropical convective heating may become less clear. One idea is to mention about the response to tropical convective heating and cite Matsuno and Gill papers, but not to say Matsuno-Gill "pattern". (Please note that the revised manuscript still has the paper by Matsuno and the one by Gill in the reference list, though these are actually not cited currently. Please go through the manuscript and make necessary corrections which are related to the substantial revisions.)

In the new version of the manuscript we have added a sentence saying that the anticyclonic dipole, apearing as a response to the equatorial anomaly of diabatic heating, is possibly associated with the excitation of an internal Rossby mode as described in Matsuno (1966) and Gill (1980) studies. (Line 372)

The choice of QBO indices, i.e. at 10 hPa and 20 hPa, is still somewhat annoying to me. Perhaps, the authors add discussion for potential reasons. It is possible that the phase relationship between zonal wind at these levels and temperature around the tropopause would be relevant. Also, if not only local dehydration but also transport process are involved, we would need to consider a further phase shift (e.g. a month or two?). These might be a potential reason why 10 or 20 hPa zonal wind results in a better correlation. Please include such discussions, hopefully quasi-quantitative ones, for this problem.

The manuscript clarifies, in the methodology section, the reasons for the choice of the 10hPa and 20hPa levels for the QBO phase definition and explains that this is directly related to the fact that the phase definition at these levels maximises the signal over the water vapour in the AM. On the other hand, it is explained that it does not mean that the physical mechanism explaining the QBO signal on the AM water vapor has a direct relationship with the circulation or temperature of the QBO at these levels. However, the definition of the phase at high stratospheric levels fixes the characteristics of the QBO throughout the stratosphere including the UTLS, where the QBO wind and temperature can have an impact on lower stratospheric water vapour.

The reviewer is right in pointing out that, the QBO impact on the tropopause temperatures, other processes related to changes in vapour transport into the monsoon could also contribute to explain the observed signal. In the new version, in the Summary and Discussion section, we have included a paragraph discussing the possibility that changes caused by the QBO on the trajectories of the air masses reaching the AM, or the variations of such trajectories with the summer progress, could also explain part of the observed QBO signature on the AM water vapour or the intraseasonal evolution of this signature. We have indicated that future studies incorporating Lagrangian transport models will be necessary to address this question and determine the role of this pathway (Lines 596-602).

For other issues that I raised in the first review, it looks the authors made reasonable revisions.

Finally, I happened to notice that the caption of Figure 7 may need a correction. The figure legend says, now it is CI at 100-150 hPa, while the caption says, fraction of cloud cover averaged between 125hPa and 150hPa. This may be related to the changes for this revised version. There might be some other places where similar corrections are needed. So, again, please go through the manuscript and make necessary corrections which are related to the substantial revisions.

The caption of figure 7 has been corrected (Line 444).

We have revised the manuscript and corrected a couple of sentences to improve its understanding. (Lines 132 and 562).

---

## Author Response (AR3)

Response to editor

Dear authors,

I think you have adequately replied to the reviewer's concerns. Therefore I am happy to accept your paper for final publication in ACP subject to technical corrections as follows:
I think panel (b) and panel (c) of Figs. 5 and 7 are mixed up in the figure caption. I think the reference of the last sentence of the figure captions "Dots indicate significance at the 95% confidence level." is also mixed up (I think it belongs to the panel indicated as (c) in the figures but as (b) in the figure captions).

Please make necessary corrections and resubmit your paper for final copy-editing, type-setting, and proof-reading.

Kind regards,
Gabriele Stiller

Dear editor,

Following your indications, we have corrected the captions of figures 5 and 7.

Thanks very much,
Cristina Peña